# Regional and global climate for the mid-Pliocene using the University of Toronto version of CCSM4 and PlioMIP2 boundary conditions

Deepak Chandan[1] and W. Richard Peltier[1]

[1]Department of Physics, University of Toronto, 60 St. George Street, Toronto, Ontario, M5S1A7

*Correspondence to:* Deepak Chandan (dchandan@atmosp.physics.utoronto.ca)

**Abstract.** The Pliocene Model Intercomparison Project, Phase 2 (PlioMIP2) is an international collaboration to simulate the climate of the mid-Pliocene interglacial, marine isotope stage KM5c (3.205 Mya), using a wide selection of climate models with the objective of understanding the nature of the warming that is known to have occurred during the broader mid-Pliocene warm period. PlioMIP2 builds upon the successes of PlioMIP by shifting focus onto a specific interglacial and by using a revised set of geographic and orbital boundary conditions. In this paper, we present the details of the mid-Pliocene simulations that we have performed with a slightly modified version of the Community Climate System Model version 4 (CCSM4), and the 'enhanced' variant of the PlioMIP2 boundary conditions, and discuss the simulated climatology through comparisons to our control simulations, and to proxy reconstructions of the climate of the mid-Pliocene. With the new boundary conditions, the University of Toronto version of the CCSM4 model simulates a mid-Pliocene which is more than twice as warm as that with the boundary conditions used for PlioMIP Phase 1. The warming is more enhanced near the high-latitudes which is where most of the changes to the PlioMIP2 boundary conditions have been made. The elevated warming in the high-latitudes leads to a better match of the simulated climatology to proxy based reconstructions than possible with the previous version of the boundary conditions.

## 1   Introduction

The mid-Pliocene warm period, 3.3-3 million years ago, was the most recent time period during which the global temperature was higher than present for an interval of time longer than any of the Pleistocene interglacials. The prevalence of widespread warming during this time has been inferred from proxy based sea surface temperature (SST) reconstructions from a number of widely distributed deep-sea sedimentary cores (*Robinson et al.*, 2008; *Lawrence et al.*, 2009; *Dowsett et al.*, 2010; *Fedorov et al.*, 2012). The PRISM (Pliocene Research, Interpretation and Synoptic Mapping) version 3 (PRISM3, *Dowsett et al.*, 2010) compilation of mid-Pliocene SSTs contains data from 100 sites. In contrast, surface air temperature (SAT) estimates are more difficult to reconstruct owing to limited availability of land based proxies that are useful for palaeo-thermometry. However, some records available from high-latitudes in the northern hemisphere provide an important perspective on the magnitude of warming in that region. *Rybczynski*

*et al.* (2013) found mid-Pliocene plant fossils in peat deposits on Ellesmere Island (north of the Arctic circle in the Queen Elizabeth archipelago) which led them to estimate that the local mean annual SAT during the mid-Pliocene was $18.3 \pm 4.1°$C warmer than present while summer temperature hovered around $14°$C. *Brigham-Grette et al.* (2013) studied lacustrine records from arctic lake El'gygytgyn in NE arctic Russia on the basis of which they inferred that the summer temperature there was $\sim 8°$C warmer than today.

The warming during the mid-Pliocene is expected to have resulted in a much higher eustatic sea level (ESL) owing to the melting of large amounts of polar ice over the extended period of time during which warmer conditions prevailed. One of the earliest estimates for the mid-Pliocene ESL was provided by *Dowsett and Cronin* (1990), who derived a local estimate based upon measurements of present-day elevation above sea level of mid-Pliocene dated marine deposits in the Orangeburg Scarp, near the South Carolina and North Carolina border. After correcting the observed elevation for the regional long-term tectonic uplift rate, they arrived at a $+35 \pm 18$ m estimate of the mid-Pliocene ESL (the plus sign denotes sea level higher than present). This value was used to perform the first global reconstruction of the mid-Pliocene climate – PRISM1 (*Dowsett et al.*, 1994).

Based on an analysis of palaeoclimatic data from high-latitude deep-sea sediments, *Kennett and Hodell* (1993, 1995) argued for a mid-Pliocene sea level no more than +25 m and "probably significantly less than this for most of this interval". Their analysis led to the revising downward of the sea level estimate for PRISM2 (*Dowsett et al.*, 1999) to +25 m. Subsequent versions of the PRISM reconstruction, namely PRISM3 (*Dowsett et al.*, 2010) and PRISM4 (*Dowsett et al.*, 2016) also include a +25 m mid-Pliocene ESL. A number of other estimates for the ESL all fall in the range of +10 m to +30 m (*Wardlaw and Quinn*, 1991; *Krantz*, 1991; *Dwyer and Chandler*, 2009; *Naish and Wilson*, 2009; *Rowley et al.*, 2013). *Miller et al.* (2012) performed a multi-proxy analysis using backstripped records from Virginia, New Zealand and the Enewetak Atoll, benthic foraminiferal $\delta^{18}$O constraints and $\mathrm{Mg/Ca} - \delta^{18}$O estimates to argue that the mid-Pliocene sea level was, within a two standard deviation margin, $22 \pm 10$ m higher than present. More recently, *Winnick and Caves* (2015) have argued for a reexamination of the transfer function that is used to convert mid-Pliocene benthic $\delta^{18}$O measurements to sea level in order to correct for the changing $\delta^{18}$O content of Antarctic ice. On the basis of the revised transfer function, they estimate that the mid-Pliocene ESL was only 9-13.5 m higher than present.

Although estimates for the mid-Pliocene ESL show considerable spread, all of them strongly suggest substantial or total loss of the most vulnerable ice sheets - the Greenland Ice Sheet (GIS) and the West Antarctic Ice Sheet (WAIS). The GIS contains $\sim 7$ m of ESL equivalent ice (*Alley et al.*, 2005) and is known to have been greatly diminished under the influence of mid-Pliocene warmth (*Lunt et al.*, 2008; *Contoux et al.*, 2015). The WAIS which presently contains $\sim$5 m of ESL equivalent ice (*Fretwell et al.*, 2012) is mostly grounded below sea level and is therefore dynamically unstable to a runaway collapse (*Weertman*, 1974). Together the collapse of these two ice sheets could contribute up to 12 m to the mid-Pliocene ESL. The largest of the present-day ice sheets - the East Antarctic Ice Sheet (EAIS) is considered to be relatively stable owing to its bedrock lying mostly above sea level. However, the EAIS does contain a number of large subglacial basins the largest being the Wilkes and Aurora basins which are grounded below sea

level and are therefore susceptible to collapse similar to the WAIS. Consequently, mid-Pliocene ESL estimates above 12 m imply substantial amounts of additional melting in the most vulnerable sectors of EAIS.

It would not be particularly surprising if the higher warming and the accompanying melting of the polar ice took place in a world with much higher radiative forcing from the presence of atmospheric greenhouse gases in higher concentrations. However, proxy reconstructions for the mid-Pliocene atmosphere do not show any significant increase in greenhouse gas concentrations compared to the present-day. Various reconstructions for the atmospheric $CO_2$ concentration, such as those based on alkenone data (*Pagani et al.*, 2010; *Seki et al.*, 2010; *Badger et al.*, 2013; *Zhang et al.*, 2013a), $\delta^{13}C$ measurements in marine organic matter (*Raymo et al.*, 1996), $\delta^{11}B$ measurements (*Seki et al.*, 2010; *Bartoli et al.*, 2011; *Martínez-Botí et al.*, 2015), foraminiferal B/Ca ratio based estimates (*Tripati et al.*, 2009) and estimates from stomatal density in fossilized leaves (*Kürschner et al.*, 1996) show that during the mid-Pliocene $CO_2$ varied between $\sim 200$ ppmv and $\sim 450$ ppmv, with most measurements reported lying in the range 300-400 ppmv. A value of 400 ppmv is often used as a boundary condition in atmosphere only and coupled-climate models of the mid-Pliocene. Even if $CO_2$ concentrations during the mid-Pliocene were only equal to those characteristic of the present-day, however, significant melting of polar ice might still be reasonably expected due to the extended period of time during which this trace gas concentration was maintained.

Very little is known concerning the concentration of the very potent greenhouse gas methane, although it is expected to be at least as high as modern, if not higher, mainly due to its release from the thawing Arctic permafrost and from the breakdown of methane clathrates - both of which are expected consequences of a warm Arctic. In the absence of proxy estimates for methane concentration in the mid-Pliocene atmosphere, the modern day concentration value is usually used (for example in the context of the PlioMIP program) in climate models. The uncertainty in the methane concentration is partly compensated for in climate models by choosing a higher concentration of $CO_2$ .

The inference of warm temperatures and reduced land ice during a time period when the atmosphere was characterized by greenhouse gases concentrations not much different that present, and significantly less than projections for the future, is the primary reason that the mid-Pliocene has received considerable interest in recent decades. Since the future of our climate will be 'Pliocene-like' it is of vital importance that we understand the magnitude and distribution of the mid-Pliocene warming, the mechanisms for the warming and the consequences of the warming.

The Pliocene Model Intercomparison Project (PlioMIP; *Haywood et al.* (2011)) was organized with the aim of understanding the magnitude and mechanisms of the degree of warmth that proxies are indicating, by using a diverse selection of climate models. Phase 1 of the project (henceforth simply PlioMIP) was concerned with simulating the averaged climate of all warm interglacials in a 300,000 year 'time-slab' during the mid-Pliocene. Results from the eight models participating in PlioMIP found significant difference between the temperature reconstructions produced by the models, and between the models and data. While the models successfully simulated an overall warmer climate with some amplification of warming in high-latitudes, they fell well short of capturing the magnitude of warming at high-latitudes that is indicated by the proxy based inferences (*Dowsett et al.*, 2013).

The outcomes of PlioMIP have helped in developing the revised strategy that is being employed in the context of PlioMIP2. The focus is now on a single warm interglacial, namely marine isotope stage KM5c (3.205 Mya). This approach is expected to improve the resolution of reconstructed temperature data and thereby improve confidence in it. On the modeling side, a new set of boundary conditions - the PRISM4 boundary conditions (*Dowsett et al.*, 2016) are available for modeling groups to use. In this paper we will employ these new boundary conditions in an attempt to better understand what the characteristics of the climate of the mid-Pliocene would be expected to have been.

## 2   Model description

We will employ for the purposes of the analyses to be discussed herein the Community Climate System Model version 4 (CCSM4; *Gent et al.* (2011)) for simulating the mid-Pliocene. CCSM4 is a coupled-climate model and consists of four major components that interact with each other through a flux coupler without flux adjustment. These components are briefly described in this section. We use a slightly modified version (described in sections 2.1.1 and 3 below) of the default configuration of the model and we refer to it as the "University of Toronto version of CCSM4" to distinguish it from the publicly released version of CCSM4 that has been run in its default configuration. Henceforth, in this paper we will employ 'CCSM4' to refer to the Toronto version when it is mentioned in the context of one of our simulations, and to the default version otherwise. CCSM4 has been used in many palaeoclimate studies and it is one of the models previously employed in PlioMIP (*Rosenbloom et al.*, 2013).

### 2.1   Atmosphere

The atmospheric component in CCSM4 is the Community Atmosphere Model version 4 (CAM4; *Neale et al.* (2013)) which is the sixth generation atmospheric general circulation model developed at the National Center for Atmospheric Research (NCAR). In this version the default dynamical core has changed from a spectral core which was used in CAM3, to a finite volume (FV) dynamical core. The FV core improves tracer transport capabilities over the spectral core and also reduces the surface zonal wind bias that was present in CAM3. In coupled mode, this bias was responsible for an intensified Antarctic Circumpolar Current (ACC) and excessive transport through the Drake Passage (*Large and Danabasoglu* , 2006). CAM4 also includes changes to the parametrization of deep convection which significantly improves the Madden-Julian Oscillation (*Subramanian et al.*, 2011) and, in coupled CCSM4 mode, the EL Niño-Southern Oscillation (ENSO) variability (*Deser et al.*, 2012). All our simulations are performed on a FV grid with $\sim 1°$ horizontal resolution (192 grid cells in the latitude and 288 grid cells in the longitude) with 26 levels in the vertical.

### 2.1.1   Ocean

The ocean component in CCSM4 is the Parallel Ocean Program, version 2 (POP2; *Smith et al.* (2010)) which was developed at the Los Alamos National Laboratory. It includes significant improvements over version 1 such as the

elimination of the cold bias in the equatorial Pacific cold tongue while also increasing the sharpness of the pacific thermocline, and the reduction of the sea surface temperature (SST) and sea surface salinity (SSS) bias along the path of the North Atlantic Drift.

POP2 also comes with a new parametrization for the overflow of density-driven currents over oceanic ridges, such as the overflow of the Nordic Sea waters over the Denmark Strait and the Faroe Bank Channel, and the overflow of dense waters over bottom topography in the Ross Sea and Weddell Sea regions of the Antarctic (*Danabasoglu et al.*, 2010). These overflows are thought to be important to the formation of deep bottom waters (*Briegleb et al.*, 2009). The new parametrization improves the penetration depth of the North Atlantic Deep Water (NADW), but introduces new biases which are worse in coupled simulations compared to uncoupled simulations (*Danabasoglu et al.*, 2010).

An important caveat to be aware of when simulating palaeo climate is that this parametrization is tuned to reproduce, to the extent possible, the observations of the overturning circulation for the present-day (such as those obtained by the RAPID array, *Cunningham et al.* (2007)). As such, its impact when included in the simulation of a time period when the ocean bathymetry would have been significantly different remains unclear. Owing to concern regarding the introduction of biases stemming from this parametrization if it were applied under Pliocene conditions when bathymetric depth was significantly modified, we have chosen to disable the parametrization in our Pliocene simulations. In addition, we have also chosen to disable this parametrization for the control simulations so that when the Pliocene simulations are compared to the controls, the differences arising from the choice of the PlioMIP2 boundary conditions can be assessed correctly, unencumbered by any other differences between the control and the Pliocene simulations.

In our simulations the ocean model runs on the displaced-pole grid in which the poles of the grid are positioned over Greenland and Antarctica and therefore the ocean does not contain any grid singularities. The grid consists of 60 vertical levels, with $320 \times 384$ cells at each level. This amounts to a nominal resolution of $\sim 1°$ although the dimensions of the grid cells are not uniform throughout the oceans.

## 2.2 Land

The land component in CCSM4 is the Community Land Model version 4 (CLM4; *Lawrence et al.* (2012)). One of the chief deficiencies of version 3 of CLM was its poor representation of the hydrological cycle, which is why version 4 includes a considerably revised hydrology with changes to the surface runoff scheme, the groundwater scheme, soil sub-model hydrology, and revised parameterizations for canopy integration, canopy interception, soil evaporation and frozen soil. These changes have led to notable improvements in soil water content, increased transpiration and photosynthesis and better simulation of the annual cycle of land water storage. Another significant improvement in CLM4 over CLM3 is the addition of a fully prognostic carbon-nitrogen biogeochemical model. This sub-model is able to affect the climate through its control of the seasonal and annual vegetation phenology. The land model runs at the same resolution as the atmosphere model, CAM4, whereas the river transport model which is included with the land model runs at a resolution of $0.5°$.

## 2.3 Sea-ice

The sea ice component in CCSM4 is based on the Community Ice Code version 4 (CICE4; *Hunke and Lipscomb* (2006)) developed at the Los Alamos National Laboratory. The main improvement in CICE4 over its predecessor, the Community Sea Ice Model version 5 (CSIM5; *Briegleb et al.*, 2004), is in the shortwave radiative transfer scheme (*Briegleb and Light*, 2007) which now uses the microscopic optical properties of snow, ice and external absorbers (such as black-carbon and dust) to explicitly calculate the multiple scattering of solar radiation in sea ice using a delta-Eddington approximation for the purposes of inferring macroscopic optical properties such as albedo, and transmission. As a result, CCSM4 has a much better representation of sea ice albedo compared to CCSM3-CSIM5 in which the albedo was parametrized using the bulk properties (thickness, temperature etc.) of sea ice and snow. The new radiative transfer scheme has also allowed for the inclusion of a simple melt pond parametrization based upon the surface meltwater flux, and inclusion of the effects of aerosol deposition and cycling. *Holland et al.* (2012) have found that the effects of direct radiative forcing by melt ponds and aerosols on the Arctic sea ice is much higher than on the Antarctic sea ice. The sea ice component operates on the same grid as the ocean component. Recent analyses of the importance of these parameterization schemes in the sea ice module of the NCAR coupled model have been presented in the context of analyses of the snowball Earth phenomenon in the papers of *Yang et al.* (2012a, b).

## 3 Design of the Numerical Experiments

This section describes the boundary conditions and the initial conditions used in our simulations along with the methodology through which they have been integrated into the CCSM4 model. The simulations are referred to using the nomenclature first employed by *Lunt et al.* (2012) and subsequently adopted with modifications for PlioMIP2 by *Haywood et al.* (2016). In this notation simulations are referred to by an abbreviated form $Ex^c$ where $c$ is the concentration of atmospheric $CO_2$ in ppmv and $x$ represents boundary conditions which have been changed from the pre-industrial (PI) such that $x$ can be absent (for the case in which no boundary conditions have been modified) or it can be either or both 'o' for a change of orography and 'i' for a change of land ice configuration.

Our control and mid-Pliocene simulations were initially integrated for several hundred model years with the diapycnal diffusivity in the ocean, $\kappa$, fixed to a constant, depth-independent background value of $0.16 \, cm^2/s$, the modern day pelagic value, and then subsequent integrations of the simulations were continued with $\kappa$ fixed to the depth-dependent profile that was used in the ocean component, POP1, of the CCSM3 model (*Collins et al.*, 2006). This profile, which we will henceforth refer to simply as the 'POP1 profile', is shown in Supplementary Figure S1 and is characterized by a depth dependent hyperbolic tangent function that gives an upper-ocean $\kappa$ of $0.16 \, cm^2/s$ and a deep-ocean $\kappa$ of $1.0 \, cm^2/s$ with the smooth transition between these asymptotic values centered at a depth of 1,000 m, the approximate depth of the main thermocline. A simulation that has been performed with the POP1 profile will be denoted by appending a P to the name derived using the nomenclature mentioned earlier and will be referred to as the 'POP1 variant'.

The motivation behind the choice of these two diapycnal mixing profiles in the context of palaeoclimate simulations, instead of the more complex and spatially varying profile that is part of CCSM4 has been previously articulated by *Peltier and Vettoretti* (2014) (henceforth, PV14). Essentially, the spatially varying component of the CCSM4 mixing field includes a significant contribution from the turbulent mixing that is generated by the flow of the barotropic tide

over rough bottom topography and which is tuned in the CCSM4 model for the modern day tidal regime (determined in part by the modern day bathymetry and coastlines). During the mid-Pliocene there were important changes to the coastlines and bathymetry that would have affected regions which are hotspots of tidal mixing in the present-day such as, the Maritime Continent, East Pacific Ocean, Labrador Sea, and the Denmark Strait and Faroe Bank Channel region in the North Atlantic. In PV14 whose focus was on the Dansgaard-Oeschger oscillation phenomenon, the

CCSM4 diapycnal turbulent diffusivity scheme was similarly disabled based upon the explicit demonstration in the papers of *Griffiths and Peltier* (2008, 2009) that the barotropic tidal regime of the glacial ocean was dramatically different from modern. Additionally, the elevations of the mid-ocean ridges could have differed from the present-day by as much as 500 m (see Figure 7 in *Dowsett et al.*, 2016, comparing PRISM4 bathymetry to modern) due to the influence of dynamic topography (*Moucha et al.*, 2008) which would have also had an important impact on the

generation of turbulent mixing. It is for these reasons that we have opted to employ simplified models of the diapycnal diffusivity for the mid-Pliocene ocean, and also for the oceans in the control simulations so as to facilitate comparison with the mid-Pliocene results without introducing ambiguity associated with the choice of different mixing schemes. Furthermore, the two different choices of the diapycnal diffusivity will allow us to assess the degree of impact that this parameter has on the strength of the meridional overturning circulation under mid-Pliocene boundary conditions.

Given that one of the defining large-scale characteristic of the global oceans is the vertical variation of the turbulent diapycnal diffusivity by an order of magnitude from low values above the thermocline to high values closer to the rough ocean floor (*Waterhouse et al.*, 2014), our POP1 variant simulations will constitute our most accurate PlioMIP2 simulations and it is these simulations that should be used by other groups for intercomparison.

## 3.1 Control Experiments

We have simulated two PlioMIP2 control experiments $E^{280}$ and $E^{400}$ (and their POP1 variants $E^{280}_P$ and $E^{400}_P$) with atmospheric $CO_2$ concentrations 280 ppmv and 400 ppmv respectively. Henceforth, we will refer to the former as the PI control, and to the latter as the modern control (which resembles modern only inasmuch as the atmospheric $CO_2$ is close to modern; all other trace gases are identical to the PI control and no urban or agricultural land units are included in the land model.) Both $E^{280}$ and $E^{400}$ are started as CCSM "hybrid runs" from an existing 3,500 year PI

control simulation (called 'cesmpifv1mts') which was run with an atmospheric $CO_2$ concentration of 280.4 ppmv and initialized in a similar manner to the glacial simulations discussed in *Vettoretti and Peltier* (2013) and PV14, in which modern day temperature and salinity were assumed (*Levitus and Boyer*, 1994) and the ocean and the atmosphere were assumed to be at rest. The model was then run continuously for 3,500 years with the overflow parameterization and the tidal mixing parametrization turned off throughout the duration of the simulation.

Cesmpifv1mts is unique to this paper but very closely related to the PI control employed for the purpose of the Dansgaard-Oeschger analyses discussed in PV14 and further in *Vettoretti and Peltier* (2015, 2016). Because the PI control employed in PV14 was created by branching from an NCAR PI control simulation at year 863 (simulation b40.1850.track1.1deg.006 run in the default CCSM4 configuration), and run for an additional 1,200 years with the overflow parameterization and the tidal mixing parametrization both turned off, we were concerned that this PI control could have inherited memory of branching from the standard configuration of CCSM4. We therefore considered it wise to produce cesmpifv1mts as an entirely new PI control. In retrospect this PI control is found to be very close to that employed in PV14. Timeseries for the evolution of the globally averaged ocean temperature, SST, salinity, and Arctic sea ice area and volume in cesmpifv1mts are shown in the Supplementary Materials section 2. In the last 500 years of this simulation, the global ocean temperature was drifting at a rate of only $0.01\,^\circ\mathrm{C/century}$. Another slight difference between cesmpifv1mts and the PI control in PV14 (a technical difference essentially irrelevant to the simulated climate) is that the modern-day topography and bathymetry for cesmpifv1mts was generated in-house from ETOPO1, whereas the PI control in PV14 used the default NCAR generated modern-day boundary conditions based upon ETOPO2v2, because that simulation was branched from an NCAR PI control.

Therefore our two PlioMIP2 control experiments $E^{280}$ and $E^{400}$ have a common ancestor (cesmpifv1mts) and they have identical orography, bathymetry, land ice and river directions. The vegetation, soil and wetland/lakes distribution in the land model in these two controls is the same as that which is employed in the 1850 configuration in CCSM4 (component set label B_1850_CN). The concentrations of the atmospheric trace gases and orbital parameters are set to those that are prescribed for PlioMIP2 and are listed in Table 1. Supplementary Materials section 3 presents a comparison of the PI SST and sea ice concentration between the POP1 variant of our control ($E^{280}_P$) and the HadISST dataset (*Rayner et al.*, 2003).

## 3.2 Pliocene Experiments

We report on two PlioMIP2 mid-Pliocene experiments namely the Core simulation $Eoi^{400}$ and the Tier 1 simulation $Eoi^{450}$. In keeping with the notation discussed above, both these simulations include the modified orography (and bathymetry) and land ice specified by the PLioMIP2 "enhanced" boundary conditions described in *Dowsett et al.* (2016). The POP1 variants of these simulations have also been performed. Additional PlioMIP2 mid-Pliocene experiments, and experiments (not included in the PlioMIP2 experiment list) exploring the sensitivity of the climatology to variations in PlioMIP2 boundary conditions were also performed, but the results of these simulations will not be discussed in the present paper. In order to keep our discussion concise and straightforward, we will focus only on the 400 ppmv Pliocene simulations ($Eoi^{400}$ and $Eoi^{400}_P$) and will refer to results from the 450 ppmv simulations only in selected discussions. The following sub-sections describe the process through which the PlioMIP2 enhanced boundary conditions have been integrated into the CCSM4 model.

### 3.2.1 Orography and the Atmosphere Model

The orography, or above sea level topography which includes the topography of ice sheets, appears as a boundary condition primarily in the atmospheric component of the coupled model where this field is ingested by the model in the form of the surface geopotential height field. Another application of orography is in the generation of the global dataset of river direction vectors which is used by the land model and is discussed further below. Following the anomaly method (*Haywood et al.*, 2016), we have first computed the anomaly between the PlioMIP2 mid-Pliocene orography and the PlioMIP2 modern orography and then superimposed this anomaly, shown in Figure 1, on our local modern orography. This method leads to subtle differences between the resulting land-sea mask (LSM) and the LSM prescribed in PlioMIP2 and therefore in an effort to maintain the integrity of the PlioMIP2 LSM, we conditionally overwrite the orography of any grid cell where the LSM differs from that described by PlioMIP2.

### 3.2.2 Bathymetry and the Ocean Model

The vertical levels in the POP2 ocean model are referred to as "KMT levels" whose values are in the range 1-60. Prescribing the mid-Pliocene bathymetry thus requires generating a two-dimensional array containing the KMT levels of each ocean grid cell. An initial KMT field is easily obtained by using the POP2 model's ancillary scripts and the PlioMIP2 mid-Pliocene bathymetry. However, this raises a number of issues which need to be addressed. These issues fall into two categories, (i) conflict between the LSM on the displaced dipole ocean grid and the PlioMIP2 LSM, particularly along straits and regions that contain archipelagos or complex coastlines, and (ii) issues pertaining to the requirements of the ocean model such as the minimum KMT level, and the minimum width of narrow straits, in terms of the number of grid cells, which is required to permit flow through them.

We have created a suite of graphical tools to enable the targeted editing of boundary condition data products for CCSM4, such as editing the KMT levels. Figure 2 shows a screen-shot from one of the tools - KMTEditor which is seen here zoomed over a region of the North Atlantic. The thick black rectangle is a cursor over a pixel and the pixels with thin black borders indicate those cells whose values have been edited. The ocean model is initialized from a state of rest with modern day temperature and salinity derived from *Levitus and Boyer* (1994). No adjustment was made to the global salinity for the +25 m mid-Pliocene sea level because any applicable adjustment would be negligible. In comparison, for the case of the Last Glacial Maximum, at which time the globally averaged sea level was lower than present by approximately 120 m, coupled-climate simulations increase the ocean salinity by 1 PSU compared to modern to account for the reduced volume of the oceans (see *Vettoretti and Peltier*, 2013; *Peltier and Vettoretti*, 2014).

### 3.2.3 Land Model and River Transport Model

The land model is capable of dynamically predicting vegetation types as a response to evolving local climate. However, the simulations described here are run with the dynamical vegetation scheme turned off. Instead, we prescribe the mid-Pliocene vegetation reconstruction by *Salzmann et al.* (2008) which was also used in PlioMIP. This reconstruction

is generated using the BIOME4 scheme (*Prentice et al.*, 1992) and is available either as a 28-type 'biome' dataset or a 9-type 'mega-biome' dataset. Because of the uncertainties that are associated with vegetation reconstructions so far back in time, we use the mega-biome dataset instead of the biome dataset so as to minimize error.

The CLM4 model represents vegetation in terms of 17 plant functional types (PFTs) and is able to represent multiple PFTs and land units within a grid cell using a sub-grid hierarchy. In order to be able to use the *Salzmann et al.* (2008) reconstruction, the mega-biome types have to be mapped into PFTs. We have generated one such mapping by projecting the modern day mega-biome types onto the modern day PFTs. Such a map turns out to be a one-to-many map but it is readily accommodated in CLM4 because of its ability to represent multiple PFTs per grid cell. The detailed properties of this mapping are described in the Supplementary Materials.

The soil color was kept fixed to modern (*Lawrence and Chase*, 2007). The direction vectors for river routing were generated from the gradient of the topography. These were inspected and edited manually using another graphical tool to ensure that all rivers reach the oceans (or an inland sea) and that there are no river loops which would prevent some fresh water from reaching the oceans and consequently leading to salinity drift in the ocean.

## 4   Results and Discussion

We have organized the results of our numerical experiments into seven individual sub-sections. In each section the discussion is based upon the climatology that is simulated for the mid-Pliocene in comparison to that of the control experiments. These climatologies represent the average over 30 model years (averaging years indicated in Table 2) which is the required averaging duration that was agreed upon at the PlioMIP2 workshop in Leeds in 2016. We mentioned earlier that the POP1 variant simulations will be our primary PlioMIP2 simulations in view of the more physically appropriate form of the diapycnal diffusivity profile employed for these simulations. Results for the constant kappa profile will be discussed only for comparison purposes when we wish to discuss the sensitivity of our results to variation of the diapycnal diffusivity.

### 4.1   Model Evolution

The objective of PlioMIP2 is to simulate the equilibrium climate of a typical interglacial in the mid-Pliocene. Therefore, the climate simulations need to be run for a length of time that is sufficiently long to ensure that the various components of the coupled-model system have come into equilibrium, in particular, in order to ensure that the ocean, which takes much longer than the atmosphere to equilibrate, has also reached an equilibrium state. Figure 3 and Figure 4 show the evolution of the ocean temperature in our simulations, and Figure 5 shows the evolution of the air temperature in the atmospheric layer closest to the surface. In these figures, a "fork" in a timeseries denotes the branching of the POP1 variant of that experiment (see Section 3) from the constant $\kappa$ variant. In all cases the original simulations were also allowed to evolve for considerable time. The total number of model years for which each of our simulations has been run, and the top of the atmosphere (TOA) energy imbalance over the climatology

are listed in Table 2. All simulations have very low TOA energy imbalance indicating that the models are in very close to statistical equilibrium states.

It is observed that while the SSTs (Figure 3a) in all models have come into equilibrium (Table 3), the deep ocean is continuing to warm (Figure 3d). It is also seen that introduction of the POP1 profile for $\kappa$ leads to greater exchange of heat between the upper ocean and the deeper ocean as expected on physical grounds. This increases the rate of warming of the deeper ocean and decreases the rate of warming for the upper and the middle ocean. The ocean as a whole continues to take up heat in all models (Figure 4), although the trends are quite small (Table 3). The ocean in the PI control shows the slowest rate of warming ($0.03\,^{\circ}\mathrm{C}/\mathrm{century}$), even after the introduction of the POP1 profile for $\kappa$, which is due to the fact that this control run was initialized from an existing, well equilibrated (Supplementary Figure S3) 3,500 year control and integrated for a further 1,700 years which has given the deep ocean sufficient time to come into equilibrium. The ocean in the modern control is warming at a slightly faster rate of $0.06\,^{\circ}\mathrm{C}/\mathrm{century}$. The oceans in the $\mathrm{E}oi^{400}\mathrm{P}$ and $\mathrm{E}oi^{450}\mathrm{P}$ mid-Pliocene simulations are warming at rates of $0.05\,^{\circ}\mathrm{C}/\mathrm{century}$ and $0.05\,^{\circ}\mathrm{C}/\mathrm{century}$ respectively at the end of over 2,500 years of integration.

## 4.2 Surface Air Temperature

The mean annual surface air temperature (MASAT) anomalies (i.e. anomalies computed using 2 meter air temperature) of the $\mathrm{E}oi^{400}\mathrm{P}$ mid-Pliocene simulation compared to both our control simulations are shown in Figure 6. A polar stereoscopic projection is employed for these graphics because the MASAT anomalies exhibit distinct spatial features only in the high latitudes. Compared to the PI, the mid-Pliocene temperatures at both poles are much higher (Figure 6a-b). The pattern in the northern hemisphere bears resemblance to the polar amplification that is observed in contemporary measurements of surface temperatures. However, the mid-Pliocene differs from PI (and modern) in the land-sea mask, ice sheet configuration and vegetation. Because of these changes, the amplification near the poles during the mid-Pliocene is expected to include feedback mechanisms which are not active under present-day conditions. The significant warming of the mid-Pliocene SATs over the high-latitude North Atlantic, Arctic and the Labrador Sea compared to PI, as shown in Figure 6a, is also in part due to the significant difference in the sea ice concentration between the mid-Pliocene and the PI control (section 4.7).

The MASAT anomaly compared to modern is shown in Figure 6c-d. Because both the mid-Pliocene and the modern experiments have the same atmospheric $CO_2$ concentration, this anomaly to a good approximation, excludes the warming signal that would arise from differences in atmospheric $CO_2$, such as was the case for the MASAT anomaly compared to PI conditions. Therefore the MASAT anomaly with respect to modern conditions represents the impacts of other changes related to the mid-Pliocene boundary conditions and the feedbacks associated with those changes. Naturally, the amplitude of warming is expected to be reduced compared to what would be obtained in comparison to PI conditions. The zonal averages of both MASAT anomalies (solid lines in Figure 8) demonstrate that the anomaly compared to modern is about $1\,^{\circ}\mathrm{C}$ lower than the anomaly compared to PI throughout the tropics and the sub-tropics. However, at high northern latitudes, we begin to see a difference between the two anomalies - while in the southern

hemisphere the difference between the anomalies as compared to both control simulations continues to remain at a level of approximately $1\,^\circ$C, peaking at $\sim 2\,^\circ$C around $70\,^\circ$S, in the northern hemisphere the difference between the two anomalies diverges rapidly. The larger anomaly with respect to the PI appears to be due to the 120 ppmv $CO_2$ difference between the mid-Pliocene and PI and to the dramatic difference between the northern hemisphere LSM.

5   Sensitivity of the simulated SAT anomalies to the choice of the ocean diapycnal diffusivity is assessed by repeating this analysis with the set of simulations performed with the constant diapycnal diffusivity. Figure 8 shows the MASAT anomalies (dashed lines) of the E$oi^{400}$ simulation with respect to the PI control (E$^{280}$ in this case) and the modern control (E$^{400}$) alongside the corresponding anomalies obtained (and discussed above) within the set of simulations performed with the POP1 diffusivity profile. This comparison shows an asymmetric response of the anomalies 10   between the two hemispheres. The anomalies obtained with constant $\kappa$ simulations (dashed lines) are appreciably reduced in the southern hemisphere, compared to the anomalies obtained with POP1 type simulations (solid lines), while the opposite is true for the northern hemisphere. There are no appreciable differences throughout the tropical and extra-tropical latitudes. We believe this analysis provides a useful first estimate for the community regarding the magnitudes of changes that could be expected, and the regions where those changes could occur due to changes in 15   $\kappa$. This would assist in better understanding the differences between results from the various models participating in PlioMIP2. Furthermore, the differences in surface air temperature at high latitudes (originating from the choice of $\kappa$) are certainly going to be an important consideration for any future study, the goal of which, is to simulate the response of high-latitude mid-Pliocene ice sheets using ice sheet coupled models forced by simulated climate as boundary conditions.

20   Table 4 compares the MASAT simulated in the E$oi^{400}$P experiment to the PRISM3 interval (3.3 - 3.0 Mya) MASATs, which were used in the original PlioMIP program to compare the simulated temperatures to proxy based inferences of terrestrial temperatures (*Salzmann et al.*, 2013). A caveat that must be kept in mind when comparing our simulated results to this compilation is that the PRISM3 proxy estimate reflects an average over the long PRISM3 time interval relevant to PlioMIP, whereas for PlioMIP2 the focus has shifted onto a specific interglacial. Subject to this caveat, it 25   is observed that except for the two Siberian sites (Chara Basin and Lake Baikal) our simulated MASATs are in very good agreement with proxy inferences.

The globally averaged MASAT is $16.8\,^\circ$C for model E$oi^{400}$P and $17.3\,^\circ$C for model E$oi^{450}$P (see Table 5). The 400 ppmv mid-Pliocene simulation is $3.8\,^\circ$C warmer than the PI and $1.8\,^\circ$C warmer than the modern control, while the 450 ppmv mid-Pliocene simulation is $4.3\,^\circ$C warmer than PI and $2.3\,^\circ$C warmer than modern control. For the set of 30   simulations performed with a constant diapycnal diffusivity, the 400 ppmv mid-Pliocene simulation is $3.5\,^\circ$C warmer than the PI and $1.5\,^\circ$C warmer than the modern control. Therefore, irrespective of the choice of the ocean diapycnal diffusivity, the new PlioMIP2 boundary conditions simulate a mid-Pliocene whose globally averaged MASAT anomaly with respect to PI control is larger than the magnitude of the anomaly predicted by every model that has been exercised previously in the context of the original PlioMIP program (*Haywood et al.*, 2013) and double the anomaly 35   that *Rosenbloom et al.* (2013) found ($1.86\,^\circ$C) using CCSM4.

Our simulated anomaly is also much higher than that found by *Kamae et al.* (2016) $(2.4°C)$ using the PlioMIP2 boundary conditions and the MRI-CGGM2.3 coupled climate model. A likely explanation as to why their anomaly is lower than ours could be their choice of the 'standard' boundary conditions set which does not require changes to the land-sea mask in the model. A more likely explanation of the difference, however, is simply that the relatively
short integration length of 500 years would not have been sufficient to enable the ocean in their model to reach a state of quasi-equilibrium.

The Earth System Sensitivity (ESS) of the planet calculated from our mid-Pliocene simulations is also provided in Table 5. The ESS inferred on the basis of our 400 ppmv mid-Pliocene experiment is $7.4°C$ per doubling of $CO_2$ . Although this number is fairly high, it agrees well with the $7.1 \pm 1.0 - 9.7 \pm 1.3$ range of estimates that *Pagani et al.*
(2010) inferred using proxy based methods. Our estimate for the ESS is double that obtained using CCSM4 for PlioMIP of $3.51°C$, and significantly higher than the PlioMIP multi-model mean of $5.01°C$.

Both of our mid-Pliocene simulations have the greatest warming occurring during the months of JJA and the least warming during the months of DJF (Table 6). The warming during JJA is more than a degree greater than that during DJF for both mid-Pliocene models and compared to both of our controls. In fact, while the JJA-DJF temperature
difference in our control simulations is $\sim 3.5°C$, the difference is $\sim 4.7°C$ in the mid-Pliocene simulations. This represents an increase in seasonality in the mid-Pliocene over that under either PI or modern conditions.

In Figure 7 we show the northern hemisphere SAT anomalies in $Eoi^{400}P$ for JJA and DJF which captures this seasonality. The first row shows the anomalies relative to the PI and the second row shows the anomalies relative to modern. The increase in seasonality in the mid-Pliocene is readily apparent, especially in comparison to modern
conditions such that the Pliocene winter (summer) is colder (warmer) than modern over land. The significant and widespread cooling of the northern hemisphere in the mid-Pliocene winter compared to modern is not unlike the temperature trend that has been observed in recent decades (*Sun et al.*, 2016; *Cohen et al.*, 2013; *Overland et al.*, 2011). The trends for the recent decades show that while the Arctic has been warming during the winter, large portions of Eurasia and North America have experienced a cooling trend. A large body of work has been undertaken
in an attempt to understand this (see a recent review by *Vihma* (2014)). One of the most promising explanations for the observed colling over land has come in the form of the dynamical connection that has been proposed between the Arctic surface warming, as a result of the September-October-November (SON) Arctic sea ice loss, and the corresponding equatorward shift of the jet stream which leads to greater intrusion of cold arctic air over the mid-latitude landmasses (*Overland et al.*, 2016; *Barnes and Screen*, 2015; *Deser et al.*, 2015; *Cohen et al.*, 2014). An
equatorward shift in the jet streams has been shown to occur in a dry dynamical core model from an imposed surface warming in the Arctic (mimicking the warming from sea ice loss) by *Butler et al.* (2010, also see references therein). More recently, *Deser et al.* (2015) have applied the CCSM4 model to the analysis of the atmospheric response to Arctic surface warming from sea ice loss by imposing an artificial long-wave forcing upon the sea ice component of the model to simulate sea ice loss. They found that in response to the SON Arctic sea ice loss, the CCSM4 model
simulates an equatorward shift of the jet stream.

Similarly, we find that the largest sea ice loss in the 400 ppmv mid-Pliocene simulation compared to the modern control occurs in SON (Supplementary Figure S9). The reduced sea ice coverage in our mid-Pliocene simulation leads to an increased heat flux into the atmosphere during the winter season (when the sea-air temperature contrast is largest) which leads to the warming of the atmospheric column above the Arctic more than over the surrounding
mid-latitudes. This in turn leads to a larger increase in geopotential height over the arctic compared to the mid-latitudes and drives the jet stream equatorward in accordance with the thermal wind relation and which brings the polar front closer to the mid-latitudes resulting in the cooling that is seen in Figure 7c. Figure 9 shows the boreal winter zonal-mean zonal wind anomaly between the 400 ppmv mid-Pliocene simulation and the modern control showing this equatorward shift in the zonal wind, as well as an equatorward shift in the upper stratospheric winds.

**4.3  Precipitation**

The anomaly of the mid-Pliocene annual precipitation compared to our two control simulations is shown in Figure 10. We see the presence of a strong double ITCZ which is a recognized problem with CAM4 (*Gent et al.*, 2011). The anomaly is larger with respect to the PI than it is with respect to modern conditions. This is expected as the Pliocene atmosphere is much warmer than in the PI owing to greater atmospheric $CO_2$, whereas the atmosphere in
the modern control has the same $CO_2$ concentration as the Pliocene. Despite the differences in the magnitude of the anomalies, the broad features remain the same. The precipitation increases over mountain belts such as the Andes, the Himalayas and the Tibetan plateau are due to the increased mid-Pliocene orography of these regions (Figure 1) and to the higher moisture content of the air rising above these mountain belts.

**4.4  Ocean Temperature**

Figure 11 shows the 400 ppmv mid-Pliocene SST as well as the SST anomalies compared to the PI control and the modern control. Our simulation is characterized by the existence of a fairly extensive expansion of the warm pool in the mid-Pliocene, a feature whose existence has been previously suggested on the basis of proxy based reconstructions of SSTs (*Brierley et al.*, 2009; *Dowsett et al.*, 2012; *Fedorov et al.*, 2012). The blue and the red contour lines in Figure 11a show the extent of $30°C$ and $31°C$ waters of the equatorial warm pool. Such warm waters are not present in
the PI control simulation in which the temperature throughout the equatorial warm pool is $1.5 - 2°C$ lower than the mid-Pliocene (Figure 11b). Our decision to keep the ocean configuration identical between the mid-Pliocene and the control simulations, as well as our decision to perform experiments with two choices of $\kappa$ have enabled us to determine (Supplementary Material section 4) that this expansion of the warm pool is solely due to the choice of PlioMIP2 boundary conditions. Warm waters are also present at high latitudes in the North Atlantic, and in the
Southern Ocean, where the SST anomalies show amplified warming compared to the rest of the ocean. The related shift in the east-west temperature gradient across the equatorial Pacific is expected to have a (perhaps significant) impact upon the ENSO process, an impact that will be discussed in detail elsewhere.

In the comparison of the mid-Pliocene SST to modern control SST (Figure 11c), a large region of negative anomalies is seen off the west coast of North American, implying that the ocean surface in the modern control is warmer than in the mid-Pliocene. This is because of an increase in the mid-Pliocene surface wind stress compared to present-day. In this region, the wind stress is responsible for forcing the prominent present-day California Current which

drives coastal upwelling that brings colder waters from below, thereby reducing the ocean surface temperature. The increased wind stress under mid-Pliocene conditions is responsible for greater coastal upwelling and reduction of SSTs (compared to modern) and the formation of the 'cold tongue' that extends from the coast.

Another way in which the mid-Pliocene ocean can be compared to our two controls is through the meridional profile of zonal-mean SST, which is shown in Figure 12 for the Atlantic-Arctic basin and the Southern-Indian-Pacific

basin. In both basins, the mid-Pliocene ocean is warmer than the PI by at least $2.5 - 3.5\,°C$, and warmer than the modern by at least $1 - 2\,°C$. The largest anomalies occur in the $45°N$-$65°N$ region of the Atlantic where most of the NADW forms. The data points in Figure 12 are estimates of the mid-Pliocene SST from the PRISM3 dataset categorized into three confidence levels (*Dowsett et al.*, 2012) - Very High, High and Medium confidence. Before we discuss the comparison to PRISM3 it needs to be noted that the PRISM3 reconstruction was generated for the original

PlioMIP program in which the aim was to simulate the average climate of the warm intervals in a 300,000 year time slab from $\sim 3.3$ to 3 millions years ago and therefore the PRISM3 boundary conditions and the SST reconstructions are an average over that time period. The PRISM3 dataset is therefore not strictly applicable to PlioMIP2 in which the focus is on a time-slice centered on the single interglacial peak at MIS KM5c. However, since the revised dataset that will be eventually applicable to PlioMIP2 is not yet available, we will be comparing our results to PRISM3.

We find that our simulated meridional SST profile is in rather good agreement with the limited number of data points that are available. The shaded region shows the range of the simulated mid-Pliocene temperatures over the specific ocean basin. A disagreement is seen near the equator in the Indo-Pacific basin where the SST estimates are $\sim 2\,°C$ lower than our simulated SSTs. In this region, proxy estimates in fact match better to the modern control than to the mid-Pliocene. However, this is in precise agreement with what *O'Brien et al.* (2014) have recently

argued regarding the inability of Mg/Ca and Alkenone based proxies (which have been used to reconstruct some of the equatorial region SSTs in the PRISM3 dataset) to capture the significantly warmer temperatures in this region. Specifically, they argue that the insensitivity of alkenone proxies to temperatures $> 29\,°C$ and the dependence of the Mg/Ca calibration to seawater Mg/Ca ratios cause both proxies to fail when it comes to recording the warmer temperature waters during the mid-Pliocene, which has lead to speculations concerning the possible operation of

'thermostat like mechanisms' that might have limited the warm pool temperatures during the Pliocene. Using the $\mathrm{TEX}_{86}$ proxy as well as a revised calibration of the Mg/Ca proxy, *O'Brien et al.* (2014) argue that the equatorial warm pool temperatures were about $2\,°C$ warmer than present-day, a suggestion with which our simulated results agree. Another region of potential disagreement between the model prediction and proxy reconstruction is the high latitude North Atlantic.

We have also compared the simulated mid-Pliocene SST anomalies with respect to PI to the large compilation of SST anomaly estimates from the PRISM3 program . Figure 13 shows the PRISM3 estimate for the mid-Pliocene SST anomalies (compared to PI) categorized into the three confidence levels mentioned above. There are roughly 100 sites in the PRISM3 dataset distributed over all ocean basins and which have been arranged as a function of latitude in the figure. The data indicates that the mid-Pliocene ocean was on average a few degrees warmer than the present-day and that the warming was particularly pronounced in the high-latitudes of the northern hemisphere. The SST anomalies at PRISM3 sites obtained from our mid-Pliocene simulations $Eoi^{400}P$ and $Eoi^{450}P$ are shown on Figure 13b in blue and green dots respectively. Both simulations are able to capture the high-latitude warming in the northern hemisphere, except at the locations of the most northern data points, whose reliability has recently been called into question in the Pliocene community. Our mid-Pliocene ocean is warmer than the data suggests in the Southern Ocean but the differences are not extreme.

Also shown on Figure 13a are the mid-Pliocene SST anomalies obtained with CCSM4 (black dots; *Rosenbloom et al.* (2013)) using the PRISM3 boundary conditions (*Sohl et al.*, 2009) for the original PlioMIP program, and the multi model mean (MMM) from PlioMIP (black squares). With the PRISM3 boundary conditions, the original CCSM4 simulation had difficulty in simulating the northern hemisphere high-latitude warming. The response of the model is very flat across all latitudes. The MMM from PlioMIP similarly did not suggest any enhanced warming in the high-latitudes. However, with the PRISM4 boundary conditions the Toronto version of the CCSM4 model is seen to very well simulate the amplification near the poles when the model is run to near statistical equilibrium (Figure 13b). Recently, *Otto-Bliesner et al.* (2017) have shown using sensitivity studies conducted with the CCSM4 model that the closure of the Bering Strait and the Canadian Arctic Archipelago – both of which are the major differences between the revised PRISM4 and the older PRISM3 paleoenvironmental reconstructions in the northern hemisphere – lead to greater warming over the North Atlantic. The authors have suggested that this is due to the inhibition of the transport of fresher waters from the Pacific to the North Atlantic, leading to a stronger Atlantic Meridional Overturning Circulation and larger northward oceanic heat transport.

## 4.5  Meridional Overturning Circulation

In Figure 14 we show the evolution of the Atlantic Meridional Overturning Circulation (AMOC) maximum in our simulations. The AMOC in all our simulations appears to have reached equilibrium. The mean state of the AMOC maximum for each simulation over the 30 year climatology is listed in Table 2. It is seen that across all simulations the AMOC is $\sim 2$ Sv stronger in the variants which have the POP1 diffusivity profile compared to the variants that employs only the background pelagic value of the diapycnal diffusivity. The dependence of the AMOC on the nature of vertical mixing is expected, and therefore it is something to keep in mind when comparing AMOCs from different models, each with their own vertical mixing schemes (*Zhang et al.*, 2013b). Additionally, the AMOC shows more variability in simulations with the POP1 mixing profile.

The strength of the AMOC in the PI control is 20 Sv and 21.5 Sv for simulations $E^{280}$ and $E^{280}_P$ respectively, and in the modern control simulations $E^{400}$ and $E^{400}_P$ the AMOC strengths are 21.9 Sv and 24.2 Sv. Our climatological estimates of the PI AMOC are in satisfactory agreement with the 17.2 Sv that has been estimated over a short time span (years 2004 - 2012) by *McCarthy et al.* (2015) using measurements obtained from the RAPID monitoring array.

Several different proxies on the basis of which it has been argued that one may infer the strength of NADW or Antarctic Bottom Water (AABW) formation have been invoked to argue for a more vigorous mid-Pliocene AMOC – sometimes called the "super conveyor" – compared to the present-day. These include arguments from comparisons of mid-Pliocene benthic $\delta^{13}C$ values in ocean basins to modern day values (*Billups et al.*, 1997; *Ravelo and Anderson*, 2000; *Raymo et al.*, 1996), measurements of Nd and Pb isotope composition recorded in ferromanganese crusts

and nodules (*Frank et al.*, 2002), oceanic carbonate dissolution history (*Frenz et al.*, 2006) and reconstructions of past marine ice sheet extents in the Ross Ice Shelf regions (*McKay et al.*, 2016). However, similar to our own findings that we will report below, coupled climate models have not been able to reproduce such invigorated AMOC in the mid-Pliocene (*Haywood and Valdes* , 2004; *Zhang et al.*, 2013b), and indeed for end-of-century which like the mid-Pliocene would correspond to a warm climate, multi-model projections show a very-high likelihood of a

reduction in the strength of the NADW cell (*Collins et al.*, 2013).

    The inability of models in the PlioMIP project to simulate an energized AMOC (*Zhang et al.*, 2013b) coincides with the additional failure of these models to simulate the northern hemisphere SST amplification that proxy records suggest (Figure 13a). It has therefore been previously argued (*Haywood et al.*, 2016; *Hu et al.*, 2015) that the closure of the Bering Strait (in PlioMIP2) could lead to larger AMOC and consequently greater oceanic heat transport to

higher latitudes in the North Atlantic. If this were to happen, then that would not only reconcile the model predictions with SST proxies, but also with the proxies that suggest an intensified AMOC. Recently, *Hu et al.* (2015) reported on the effects of the closure of the Bering Strait on AMOC strength and the meridional heat transport using CCSM3 and CCSM2 under present, 15 thousand years before present and 112 thousand years before present boundary conditions. They found that under all these conditions, i.e. regardless of the background climate state, and for both models the

closure of the Bering Strait resulted in a strengthening of the AMOC by $\sim$ 2-3 Sv.

    We find that the AMOC in our 400 ppmv and 450 ppmv mid-Pliocene simulations are almost identical and their strengths are only 2-3 Sv higher than in the PI control (Figure 14). This represents an increase of the mid-Pliocene AMOC by just $10\%$ over the PI AMOC and therefore does not lend support to the idea of a significantly intensified AMOC during the mid-Pliocene. This increase is comparable to the increase that has been estimated by *Hu et al.*

(2015) to result from the closure of the Bering Strait and would therefore lead one to speculate that the stronger AMOC in our mid-Pliocene simulations is the result of the closure of the Bering Strait. However, this argument is complicated by the fact that the AMOC in the 400 ppmv modern control, which like the PI control is characterized by an open Bering Strait, is stronger than that in the mid-Pliocene simulations. Additionally, our 400 ppmv Pliocene AMOC is weaker than that simulated with the CCSM4 model in PlioMIP (which had an open Bering Strait, *Zhang*

*et al.*, 2013b), although this could be due to the much shorter model run (550 years) on the basis of which the

PlioMIP CCSM4 diagnostic was computed. Therefore, it is presently not possible to conclude that the marginal strengthening of the AMOC seen in our mid-Pliocene simulations is due to the closing of the Bering Strait.

## 4.6 Meridional Heat Transport

The atmosphere and the ocean are together responsible for transporting the excess heat that accumulates near the
equator to the high latitudes, where this heat can be radiated to space in the form of longwave radiation, thereby helping to maintain an equilibrium climate. Under present-day conditions the maximum transport of heat poleward is $\sim 5.5\,\text{PW}$ that peaks at $30°$-$40°$ latitudes in each hemisphere (*Trenberth and Carson*, 2001). The atmosphere dominates the heat transport poleward of the sub-tropics and the peak transport of heat is $\sim 5\,\text{PW}$ at $40°$ latitudes. By comparison, the ocean carries much less heat and dominates only in the deep-tropics. The maximum heat
transported by the ocean is just under $2\,\text{PW}$. These characteristics of the present-day meridional heat transport are well represented in our control simulations (Figure 15).

Figure 15 shows that the total meridional heat transport in our $\text{E}oi^{400}\text{P}$ mid-Pliocene simulation is lower than both the PI and modern controls. The reduction in the transport of heat is seen in both the atmosphere and the ocean. The atmospheric heat transport (AHT) in all our simulations (mid-Pliocene and controls) is essentially identical
throughout the tropics, the sub-tropics and the southern hemisphere. The only notable difference arises in the mid-to-high latitudes of the northern hemisphere where the mid-Pliocene AHT is lower than both controls (Figure 15a,c). A very small difference in AHT between the mid-Pliocene and the control is also noticed close to $65°\text{S}$ which might be due to the substantial differences in topography and grounded ice sheets between the two simulations at this latitude.

The reduction in the oceanic heat transport (OHT) in the mid-Pliocene is primarily due to the reduction in OHT in the Indo-Pacific basin (Figure 15b,d). It is likely that this reduction can be attributed to the closing of the Bering Strait. *Hu et al.* (2015) have shown that closing the Bering Strait can lead to a decrease in the northward heat transport in the North Pacific by $\sim 10\% - 15\%$ compared to the case with an open Bering Strait. We find that the North Pacific OHT in our mid-Pliocene simulation (which has a closed Bering Strait) is lower than in both our control
simulations (with open Bering Strait) by $\sim 20\% - 25\%$. Although a comparison between our mid-Pliocene simulations and our control simulations is not the same as comparing two simulations with identical boundary conditions save for the differences concerning the Bering Strait, as *Hu et al.* (2015) have done, our analysis suggest that their results regarding the impact of the closing of the Bering Strait on the North Pacific OHT is a robust response of the climate system that persists even when there are other differences in boundary conditions.

We find that the southward OHT in the South Pacific during the mid-Pliocene is also reduced compared to the PI. This, however, is in contrast to the consistent increase in the southward OHT in the South Pacific that *Hu et al.* (2015) found as a consequence of the closure of the Bering Strait. This suggests to us that the difference in the atmospheric radiative forcing, and the geographical changes between the mid-Pliocene and the PI control have had an impact on the southward OHT in the South Pacific, in addition to that which would be expected from the analysis of *Hu et al.*

(2015). However, we have to be cautious concerning this comparison, as the simulated differences could also be due to the differences between the ocean components in these versions of CCSM. Both CCSM2 and CCSM3 use POP1 as their ocean model, whereas CCSM4 uses the POP2 ocean model.

The OHT in the Atlantic basin during the mid-Pliocene is only marginally higher, by $\sim 3\%$, than the PI. This increase is substantially less than the $\sim 10\%$ increase in the strength of the AMOC. The mid-Pliocene Atlantic OHT is also nearly identical to the modern control despite geographical differences between the two cases. In fact, in the latitude range $50°$N-$70°$N, the Atlantic in the modern control is transporting more heat than the Atlantic in the mid-Pliocene. This further increases the difficulty in assessing whatever impact the closed Bering Strait might be having on the mid-Pliocene climate through the reorganization of oceanic heat transport, and points to the possibility of increasing meridional heat transport in the Atlantic sector without the need for closing the Bering Strait.

Finally, we note the presence of what appears to be a local Bjerknes compensation around $65°$S latitude in our 400 ppmv mid-Pliocene simulation (Figure 15a,c). In the vicinity of this latitude, the mid-Pliocene AHT is reduced compared to both the controls. It is then left up to the ocean to transport the excess heat southward of this latitude and consequently the OHT becomes greater than that in the controls. This compensation ensures that there is no anomalous change in the net meridional heat transport near this latitude.

### 4.7 Sea-ice

The mid-Pliocene sea ice is considerably reduced during both boreal and austral winters compared to PI (Figure 16). In the northern hemisphere the greatest loss in sea ice occurs (besides the Hudson Bay, which is expected not to have been present during the mid-Pliocene) in the Labrador Sea and the Greenland and Norwegian Seas in the Atlantic sector, and in the Barents and Kara Seas in the Arctic sector. The reduction in northern hemisphere sea ice is particularly pronounced in the summer months to such an extent that the Arctic can be considered ice-free.

Sea ice is uniformly reduced along the coastlines of Antarctica during the austral winter. The largest reduction in sea ice is seen in the Weddell Sea and off the coast of Queen Maud Land and Wilkes Land. The sea ice is most concentrated in the vicinity of the mid-Pliocene Antarctic archipelago and in the region presently occupied by the Filchner-Ronne ice shelf. The presence of the archipelago has allowed for the ice that today forms in the Bellingshausen and Amundsen Seas to move closer towards the pole and therefore leads to a poleward retreat of the ice margin. In austral summer the only concentration of sea ice is in the archipelago while the rest of the coastline of Antarctica is largely ice-free.

### 5 Conclusions

In this paper we have described the implementation of the revised boundary conditions for the mid-Pliocene epoch in the CCMS4 coupled climate model and employed the new structure in the reconstruction of mid-Pliocene climate conditions. We have performed two mid-Pliocene experiments, the core experiment denoted E$oi^{400}$ and the Tier 1

experiment denoted E$oi^{450}$, along with the core control experiments E$^{280}$ and the Tier 2 control experiment E$^{400}$. In addition, we have two versions of these simulations which are differentiated by the ocean's vertical profile of mixing. The first version has a constant (pelagic) value of diapycnal diffusivity throughout the ocean. The second version has its mixing profile fixed to that used by the ocean component POP1 of the CCSM3 model. The discussions and

analysis in this paper are based on the climatology that is simulated by the second version of the experiments.

We find that the PRISM4 boundary conditions mandated in PlioMIP2 lead to greater warming in the mid-Pliocene, and in particular enhanced warming at high-latitudes compared to that inferred using the same CCSM4 model and the PRISM3 boundary conditions from the original PlioMIP. The simulated 400 ppmv mid-Pliocene climate has a global MASAT that is $3.8\,°C$ higher than the PI control and $1.8\,°C$ higher than the modern day control. These anomalies

are larger than the anomalies predicted by every model previously exercised in PlioMIP and more than double the anomaly that was obtained with CCSM4 in this context (*Haywood et al.*, 2013), demonstrating that the changes to the boundary conditions have had considerable impact on the climate, but also we expect that it is important that such integrations need to be run to statistical equilibrium. In addition, we find that globally averaged temperature difference between the seasons JJA and DJF has increased during the mid-Pliocene compared to both the PI and

the modern controls. While the JJA-DJF temperature difference in both of our controls is $\sim 3.5\,°C$ the difference increases to $\sim 4.7\,°C$ in the mid-Pliocene.

The mid-Pliocene ocean that we have simulated is characterized by (i) a fairly expansive warm pool where the temperatures are $1.5-2\,°C$ warmer than the PI, and by (ii) elevated levels of warming at high-latitudes in the Southern Ocean and the North Atlantic. The SST anomalies with respect to PI agree rather well with the proxy inferred SST

anomalies compiled for the PRISM3 reconstruction. Both the 400 ppmv and 450 ppmv mid-Pliocene simulations are able to capture the mid-to-high latitude warming that is seen in the PRISM3 dataset. The agreement between the results of our simulations with the new boundary conditions, and the PRISM3 dataset is much better than that which was possible with any of the models in the original PlioMIP (*Dowsett et al.*, 2013). The caveat to our present result is that the PRISM3 SST reconstruction was generated for the original PlioMIP program in which the aim was

to simulate the average climate of the warm intervals in a 300,000 year time slab from $\sim 3.3$ to 3 millions years ago and therefore the PRISM3 boundary conditions and the SST reconstructions are an average over that time period. The PRISM3 dataset is therefore not strictly applicable to PlioMIP2 in which the focus is on a time-slice centered on the single interglacial peak at MIS KM5c. However, the revised dataset that will be applicable to PlioMIP2 is not yet available. When the dataset has been made available, we will revisit this data-model comparison.

The larger warming that is seen in both SAT and SST in our simulations at high-latitudes is accompanied by a decrease in the net meridional heat transport compared to that in either of our controls. The reduced meridional heat transport is the result of reduction in both the atmospheric and oceanic transports. Partitioning of the OHT into contributions from the major oceanic basins shows that the northward transport of heat is greatly reduced in the Indo-Pacific basin. In the Atlantic basin, the meridional transport of heat is only marginally increased ($\sim 3\%$) compared to

the PI control, while compared to the modern control, it is either identical or in the high northern latitudes, lower

than the control. This suggests that the amplified warming at the high latitudes in the mid-Pliocene, inferred from proxies and supported by our simulations, could have more to do with the local positive feedback processes activated by the changes in geography, ice sheets and vegetation, than with the increased northward transport of heat.

Lastly, we note that our simulations do not support the case for a mid-Pliocene AMOC which is substantially stronger than in the PI control. The existence of a stronger AMOC has been argued from various proxy based inferences (*Billups et al.*, 1997; *Ravelo and Anderson*, 2000; *Raymo et al.*, 1996; *Frank et al.*, 2002; *Frenz et al.*, 2006; *McKay et al.*, 2016) and it has been considered as a possible remedy to the inability of previous climate models to simulate an amplified high-latitude mid-Pliocene warming. Our mid-Pliocene AMOC is only $\sim 10\%$ stronger than the PI AMOC and slightly weaker than the AMOC in the modern control (Table 2; Figure 14). It has been argued that the closed Bering Strait in the latest boundary conditions should lead to a stronger AMOC (*Haywood et al.*, 2016; *Hu et al.*, 2015). Although our mid-Pliocene AMOC is indeed stronger than the PI AMOC, the fact that it is also weaker than the modern control AMOC makes it difficult to support the idea that a closed Bering Strait would necessarily lead to a stronger AMOC.

*Acknowledgements.* Computations were performed on the TCS supercomputer at the SciNet HPC Consortium. SciNet is funded by: the Canada Foundation for Innovation under the auspices of Compute Canada; the Government of Ontario; Ontario Research Fund - Research Excellence; and the University of Toronto. We are grateful to Dr. Guido Vettoretti for guiding DC through the intricate process of implementing boundary conditions for palaeoclimate simulations with CESM. We are also grateful to NCAR for organizing the annual CESM tutorial, one of which was attended by DC, with partial support from NCAR. We would also like to thank Dr. Bette Otto-Bliesner for funding a short visit for DC to NCAR which allowed him to understand more about the CESM model and helped him to implement the mid-Pliocene boundary conditions. DC is also very grateful to The Centre for Global Change Science, University of Toronto, which has funded multiple trips to conferences and workshops related to the work described in the paper. We thank two anonymous reviewers and the editor for their constructive comments which have significantly improved our manuscript. The research of WR Peltier at the University of Toronto is funded by NSERC Discovery Grant A9627.

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

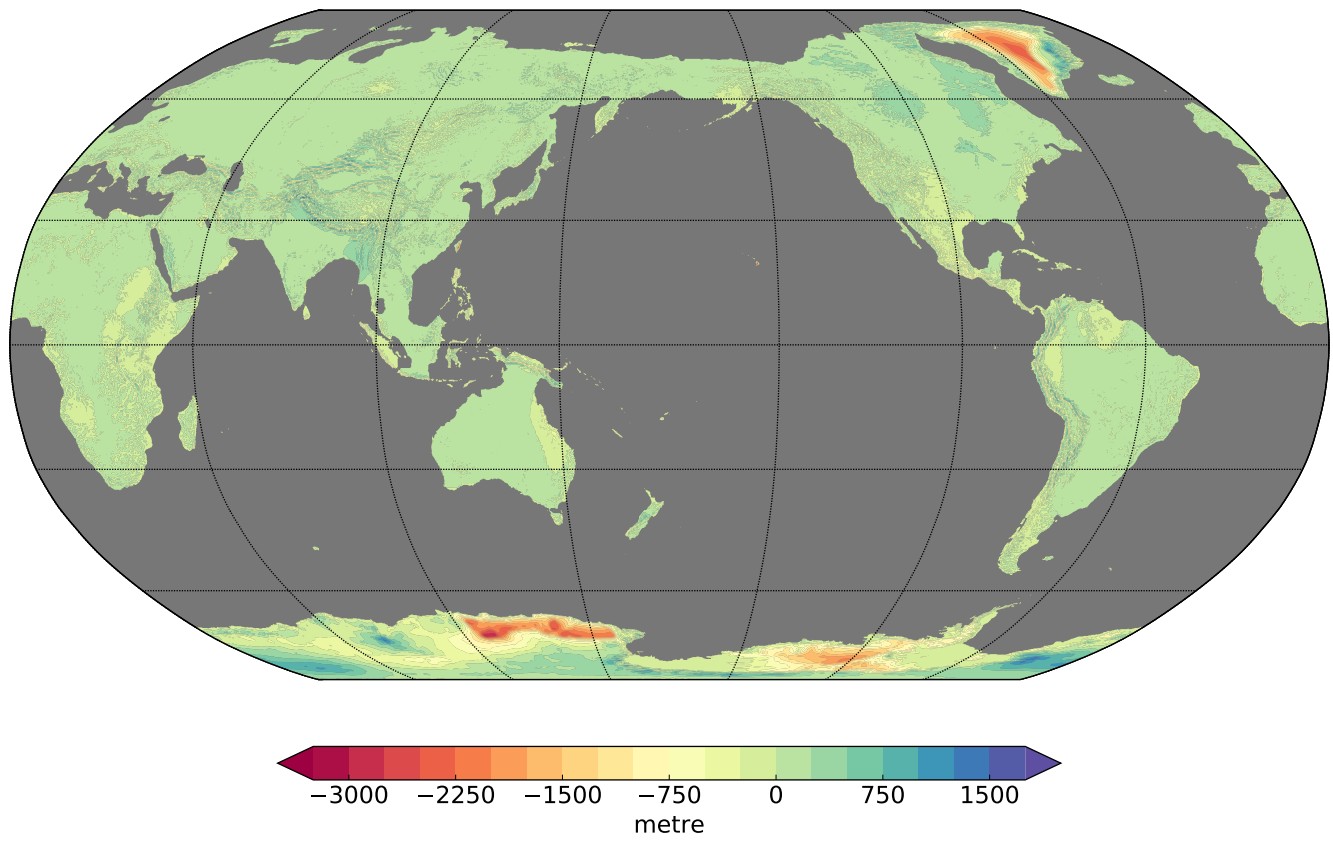

**Figure 1.** The topography anomaly field which when superimposed on our local modern day topography results in the PRISM4 mid-Pliocene topography.

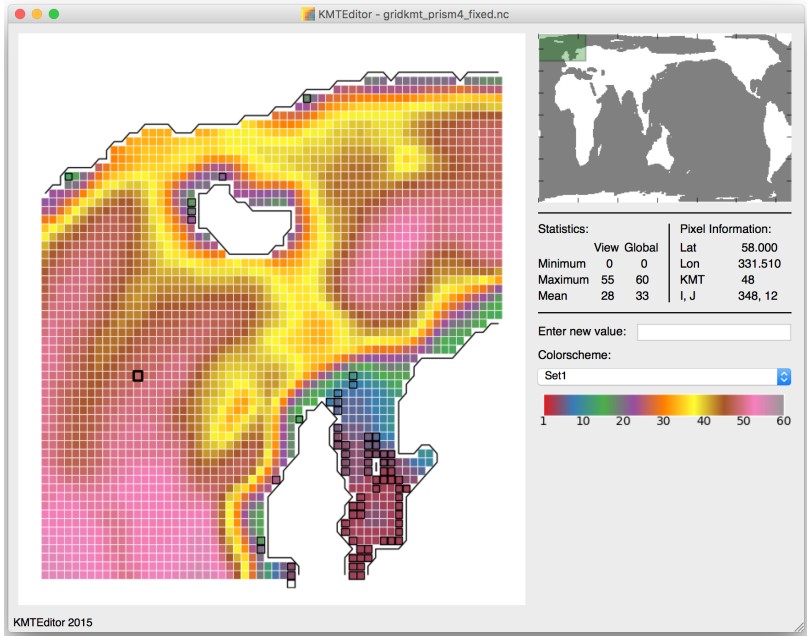

**Figure 2.** A screenshot of one of our graphical tools for editing the ocean model's grid. The dark black square in the left side plot is the cursor which can be used to navigate over to individual cells in order to change their values. The cells which are outlined by a thin dark border denote those cells whose values have been edited.

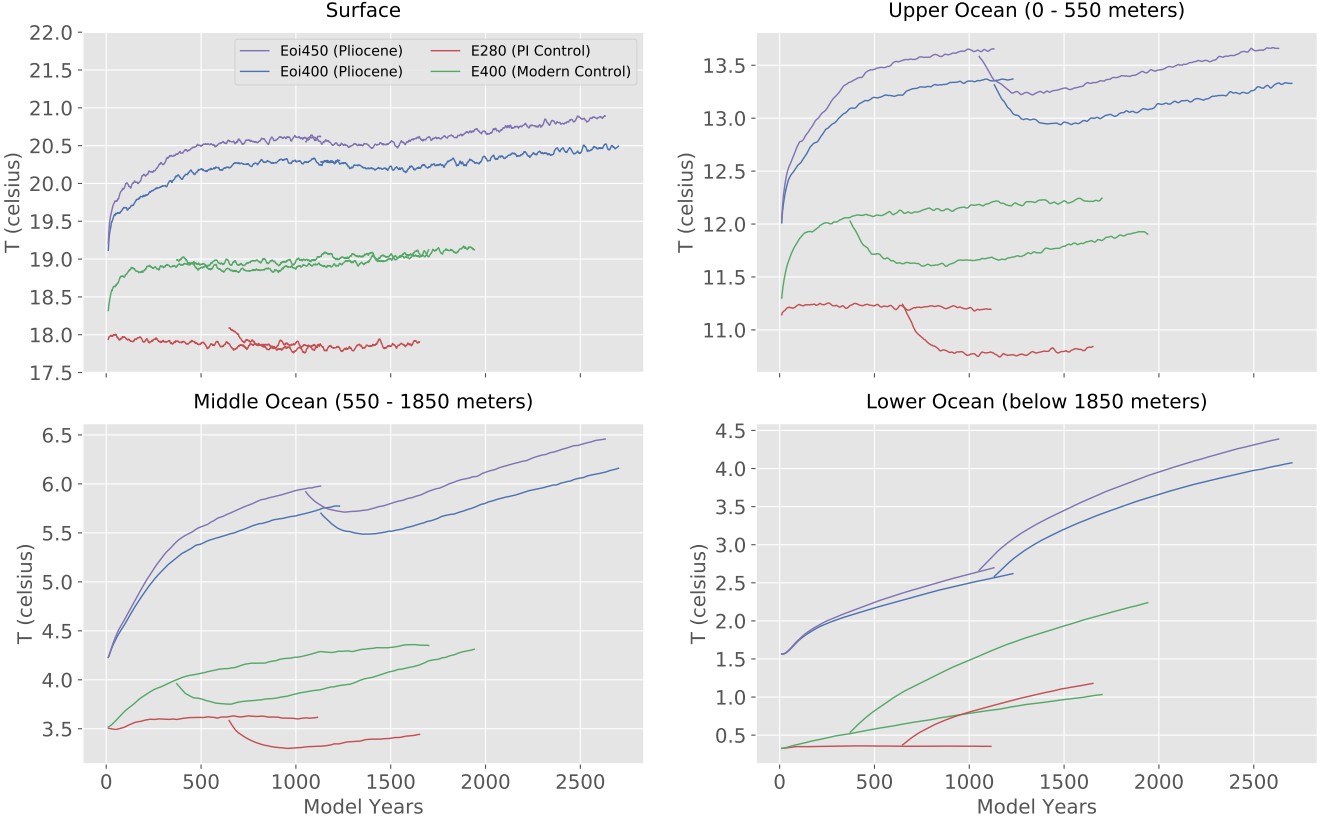

**Figure 3.** Evolution of the ocean temperature at different depths. When a new curve "forks" from an existing curve it represents the branching of the P version of that simulation in which the ocean model's $\kappa$ has been fixed to the POP1 type profile. In all cases the original simulation (with $\kappa = 0.16$ throughout the ocean) was also continued further for some more time. (a) the evolution of the sea surface temperature, (b), (c) and (d) are the volume averaged temperatures in the upper ocean (0-550 meters), middle ocean (550-1850 meters) and lower ocean (below 1850 meters) respectively.

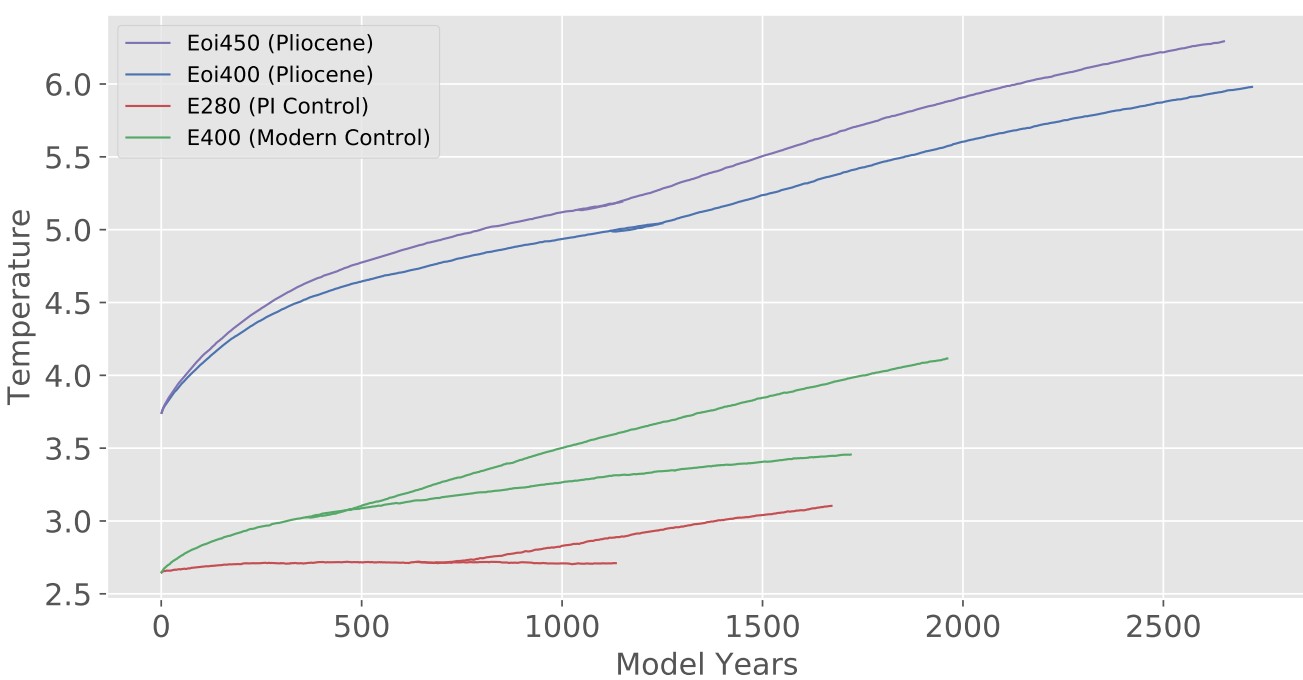

**Figure 4.** Similar to Figure 3 but instead showing the evolution of the globally averaged ocean temperature.

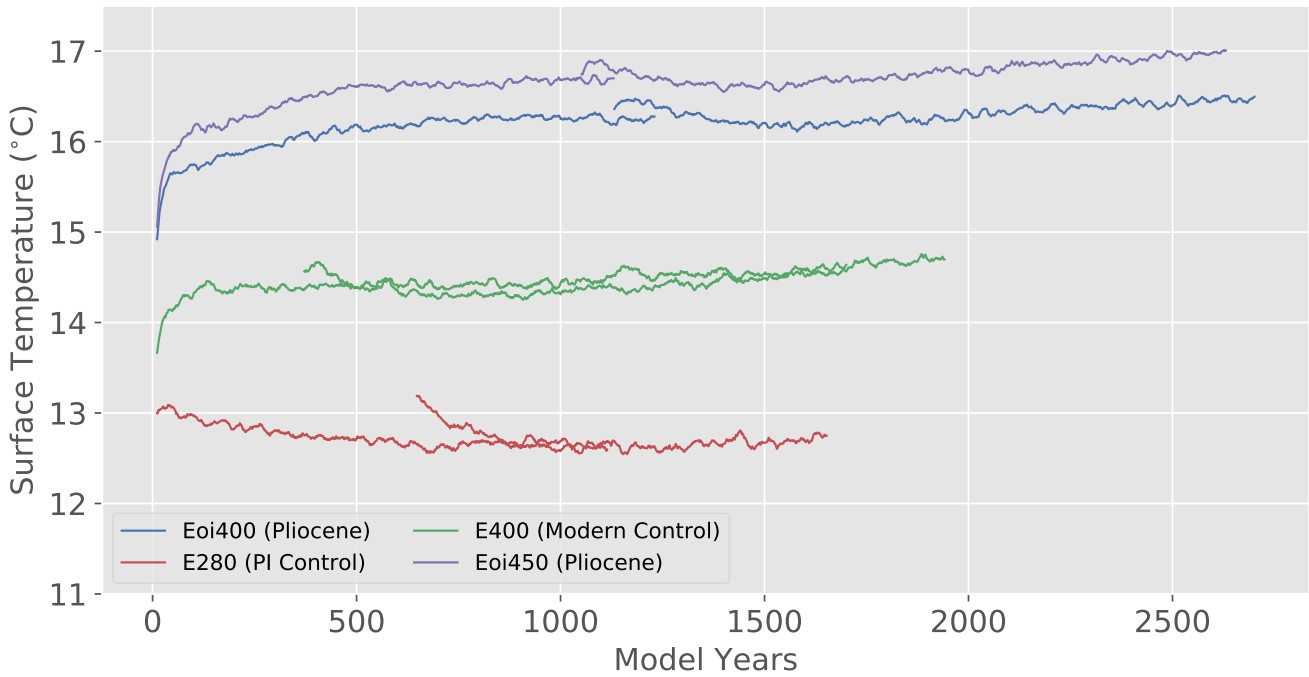

**Figure 5.** Evolution of the globally integrated air temperature of the atmospheric model layer closest to the surface. When a new curve "forks" from an existing curve it represents the branching of the P version of that simulation in which the ocean model's $\kappa$ has been fixed to the POP1 type profile. In all cases the original simulation (with $\kappa = 0.16$ throughout the ocean) was also continued further for some more time.

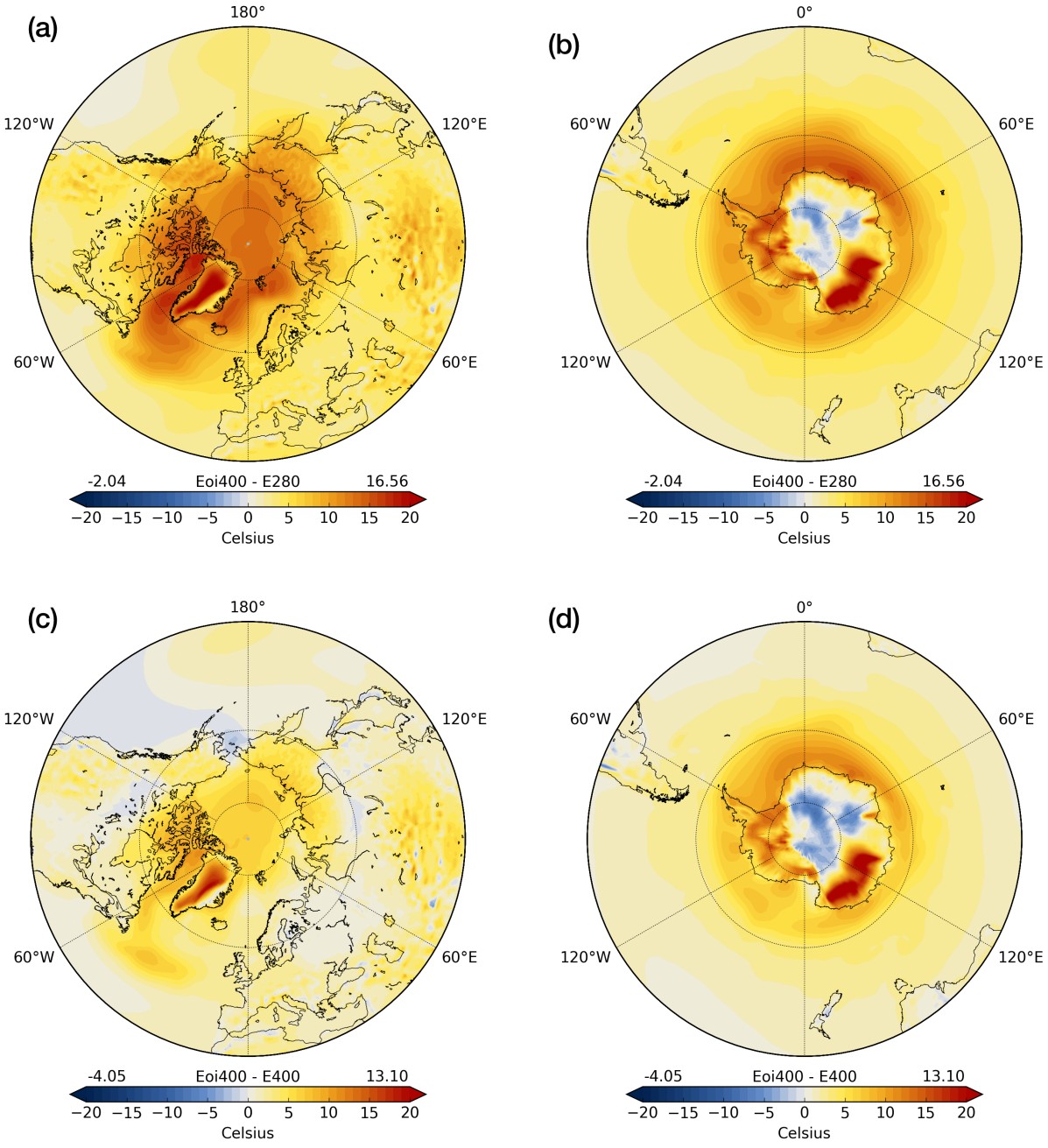

**Figure 6.** Climatological surface air temperature anomalies for the E$oi^{400}$P mid-Pliocene simulation $\langle 2691, 2720 \rangle$ compared to the pre-industrial control E$^{280}$P $\langle 5131, 5160 \rangle$ (a) and (b), and compared to the modern control E$^{400}$P $\langle 1931, 1960 \rangle$ (c) and (d). The $\langle \rangle$ denotes the range of model years over which the climatology was computed. Anomaly is defined as Pliocene - control.

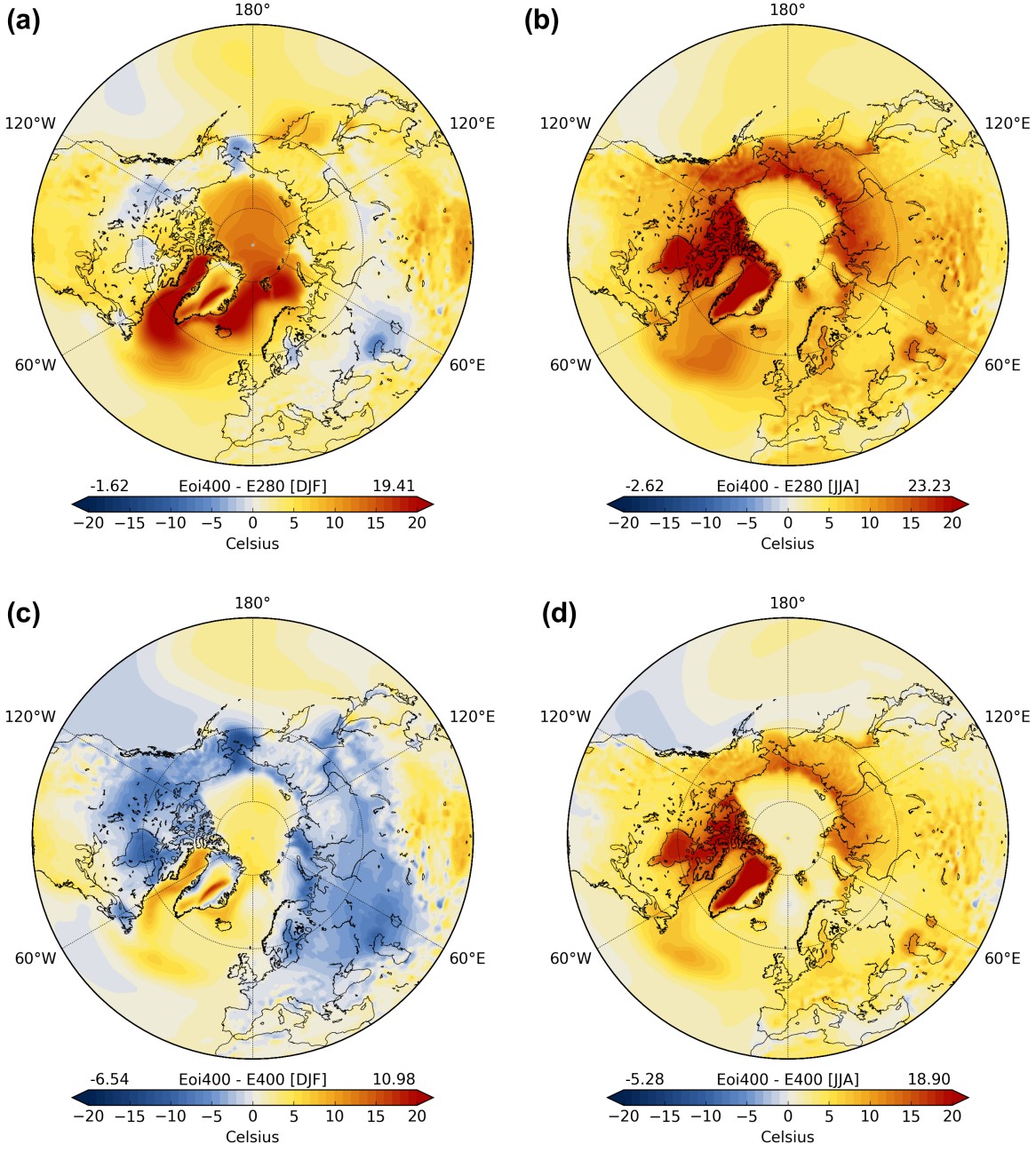

**Figure 7.** Similar to Figure 6 but instead showing the mean seasonal surface air temperature anomalies. (a) and (c) are for DJF, (b) and (d) are for JJA.

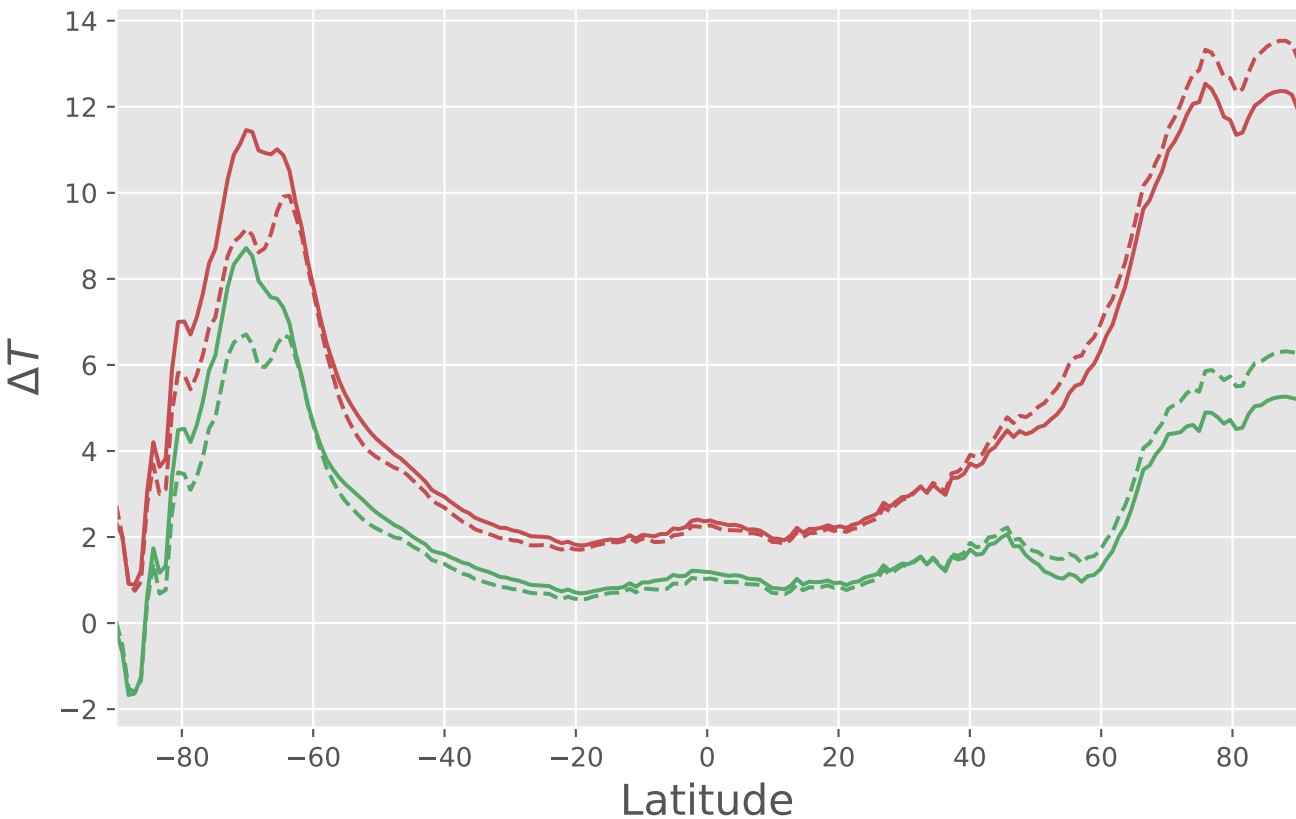

**Figure 8.** The zonally averaged climatological surface air temperature anomalies of the 400 ppmv mid-Pliocene simulation $\langle 2691, 2720 \rangle$ compared to the PI control $\langle 5131, 5160 \rangle$ (red) and the modern control $\langle 1931, 1960 \rangle$ (green). The solid lines are the anomalies computed within the set of simulations that are characterized by the POP1 profile of ocean diapycnal diffusivity and the dashed lines are the anomalies computed within the set of simulations with the constant background diapycnal diffusivity. The $\langle \rangle$ denotes the range of model years over which the climatology was computed. Anomaly is defined as Pliocene - control.

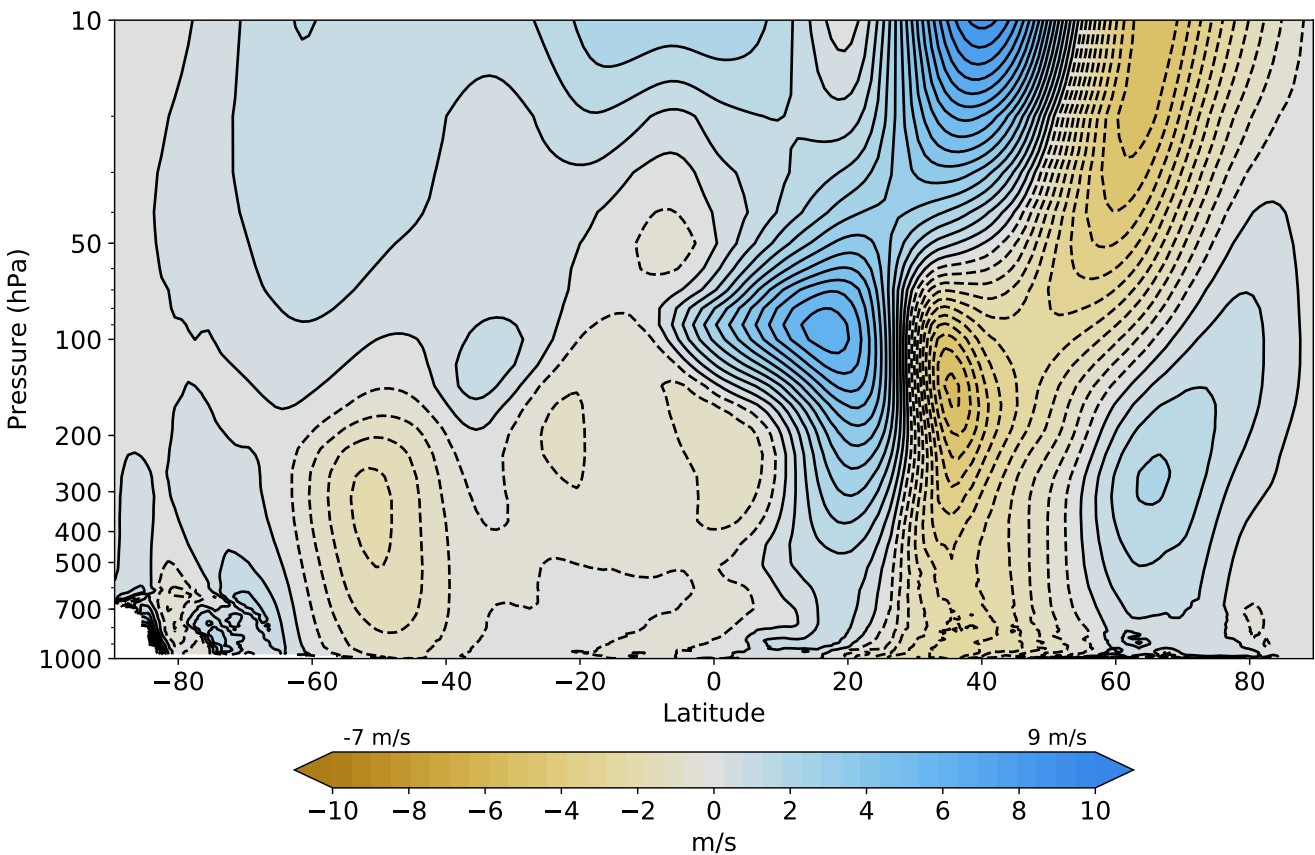

**Figure 9.** The boreal winter (DJF) climatological zonal-mean zonal wind anomaly between the 400 ppmv mid-Pliocene simulation ($\mathrm{E}oi^{400}\mathrm{P}$) $\langle 2691, 2720 \rangle$ and the modern control ($\mathrm{E}^{400}\mathrm{P}$) $\langle 1931, 1960 \rangle$ showing the prominent equatorward shift in the zonal winds over the mid-latitudes. The $\langle \rangle$ denotes the range of model years over which the climatology was computed. Anomaly is defined as Pliocene - control.

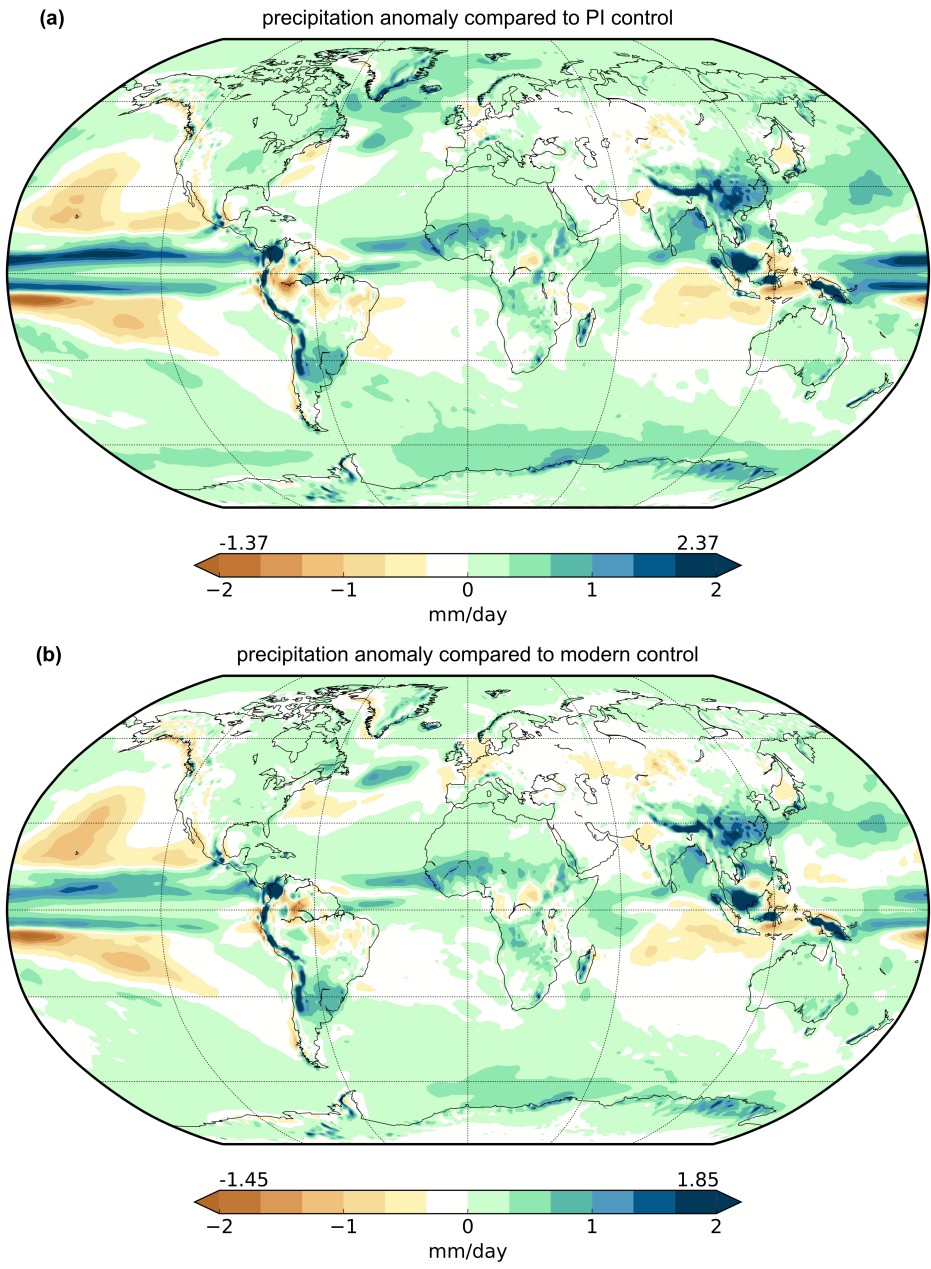

**Figure 10.** The annual precipitation anomaly between the $\mathrm{E}oi^{400}\mathrm{P}$ mid-Pliocene simulation $\langle 2691, 2720 \rangle$ and (a) the PI control $\mathrm{E}^{280}\mathrm{P}$ $\langle 5131, 5160 \rangle$ and, (b) the modern control $\mathrm{E}^{400}\mathrm{P}$ $\langle 1931, 1960 \rangle$. Anomaly is defined as Pliocene - control. The $\langle \rangle$ denotes the range of model years over which the climatology was computed.

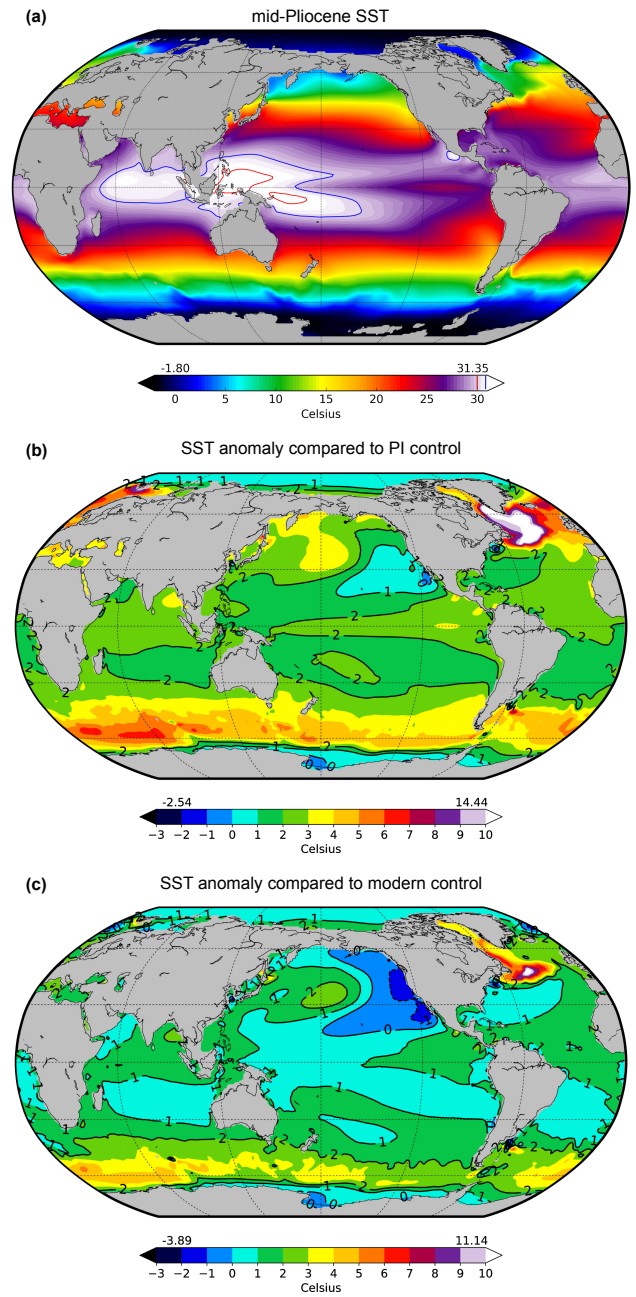

**Figure 11.** (a) The mid-Pliocene SST from the $\mathrm{E}oi^{400}{}_\mathrm{P}$ simulation $\langle 2691, 2720 \rangle$. The blue and the red contours are for $30°\mathrm{C}$ and $31°\mathrm{C}$ temperatures respectively. The SST anomaly with (b) PI control $\mathrm{E}^{280}{}_\mathrm{P}$ $\langle 5131, 5160 \rangle$ and, (c) modern control $\mathrm{E}^{400}{}_\mathrm{P}$ $\langle 1931, 1960 \rangle$. Anomaly is defined as Pliocene - control. The $\langle \rangle$ denotes the range of model years over which the climatology was computed.

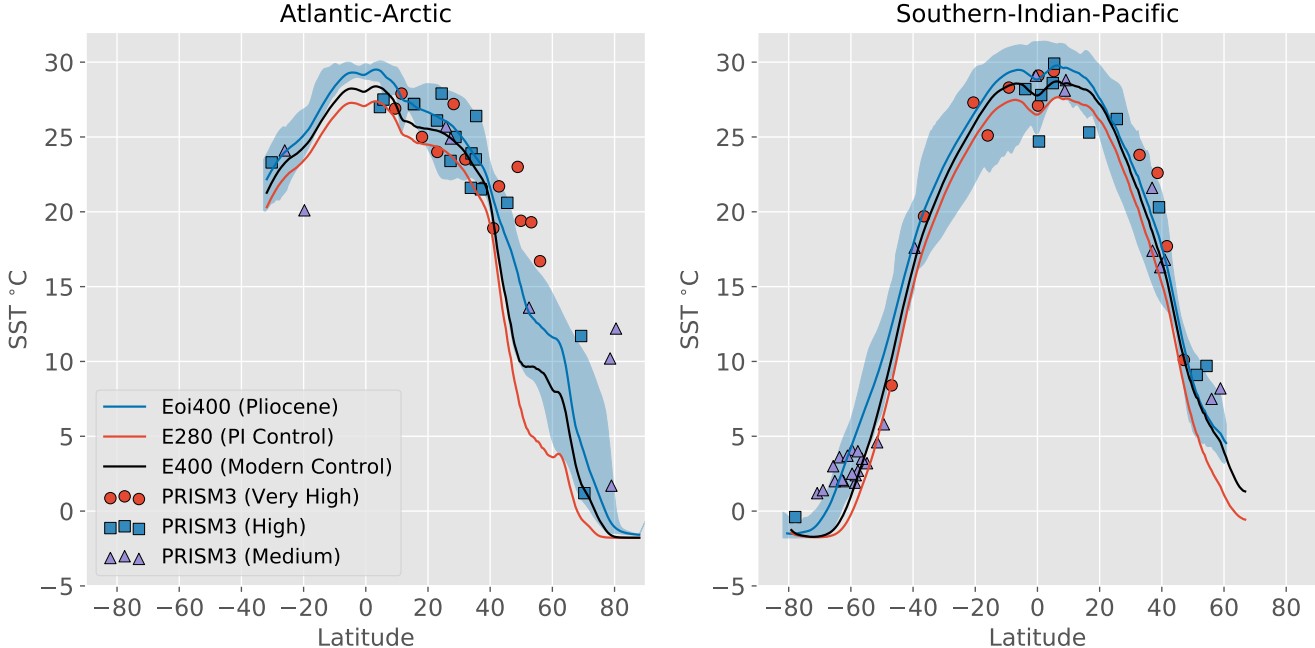

**Figure 12.** The zonal-mean SSTs for the 400 ppmv mid-Pliocene E$oi^{400}$P $\langle 2691, 2720 \rangle$, modern control E$^{400}$P $\langle 1931, 1960 \rangle$ and PI control E$^{280}$P $\langle 5131, 5160 \rangle$ for two oceanic basins. On the left is the zonal means over the Atlantic and the Arctic basins, and the zonal mean over the Southern, Indian and the Pacific Ocean is shown on the right.The data points are the PRISM3 estimates for SSTs (*Dowsett et al.*, 2010) which have been categorized into three confidence categories (*Dowsett et al.*, 2012). The shaded region highlights the range of the simulated mid-Pliocene temperature in each basin. The $\langle \rangle$ denotes the range of model years over which the climatology was computed.

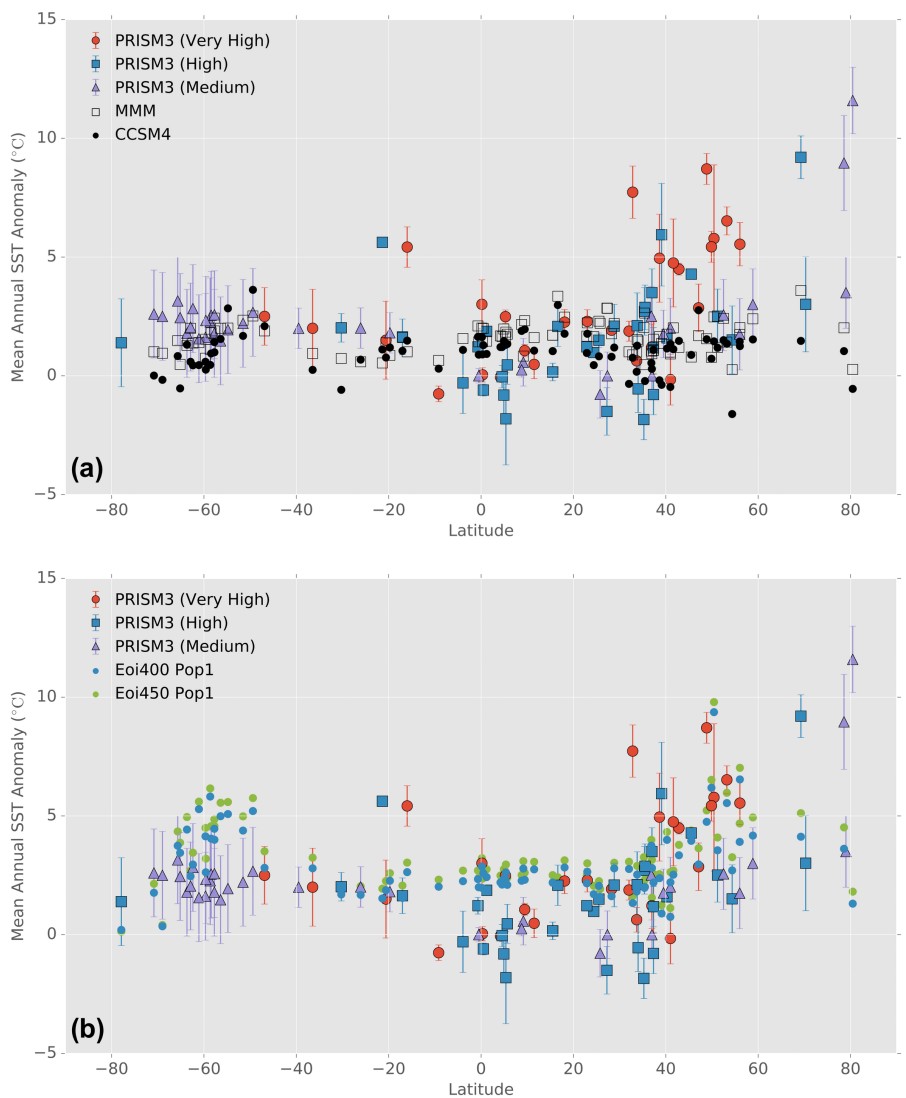

**Figure 13.** Data-model comparison of the mid-Pliocene SST anomaly compared to pre-industrial. The data in both (a) and (b) are the PRISM3 estimates (*Dowsett et al.*, 2010) which have been categorized into three confidence categories (*Dowsett et al.*, 2012). (a) compares the PRISM3 estimates to the multi-model mean from PlioMIP1 (transparent black square markers), and to the anomalies obtained by *Rosenbloom et al.* (2013) for PlioMIP1 (black dots) using the same CCSM4 model that we are using in this study. (b) compares the PRISM3 estimates to the anomalies obtained from our two mid-Pliocene simulations $\mathrm{E}oi^{400}{}_{\mathrm{P}}$ $\langle 2691, 2720 \rangle$ and $\mathrm{E}oi^{450}{}_{\mathrm{P}}$ $\langle 2621, 2650 \rangle$, shown in blue and green dots respectively. The $\langle \rangle$ denotes the range of model years over which the climatology was computed.

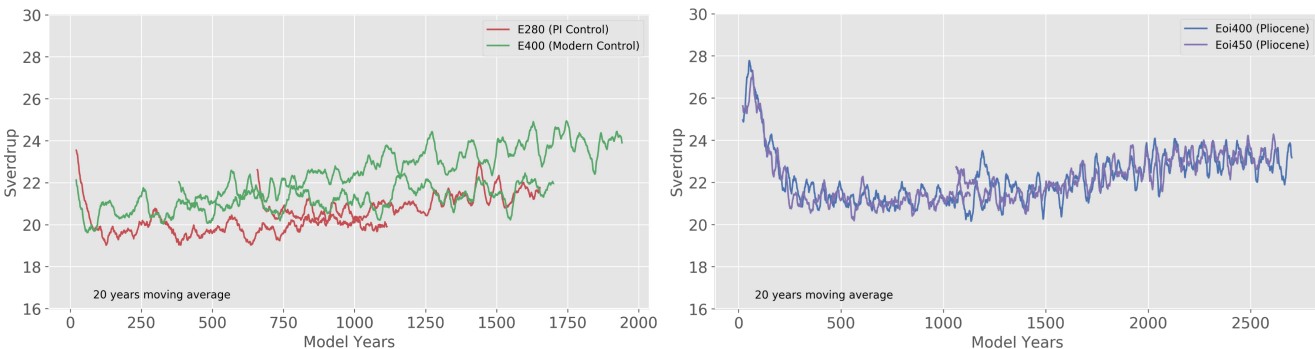

**Figure 14.** Evolution of the maximum of the Atlantic Meridional Overturning Circulation. The results shown have been filtered with a 20 year running mean to remove high-frequency variability. For each simulation the curve that starts at model year zero is for the version that uses a vertically constant background diapycnal diffusivity while the curve with identical color that starts mid-way is for the version which uses the CCSM3/POP1 type vertical profile of diapycnal diffusivity.

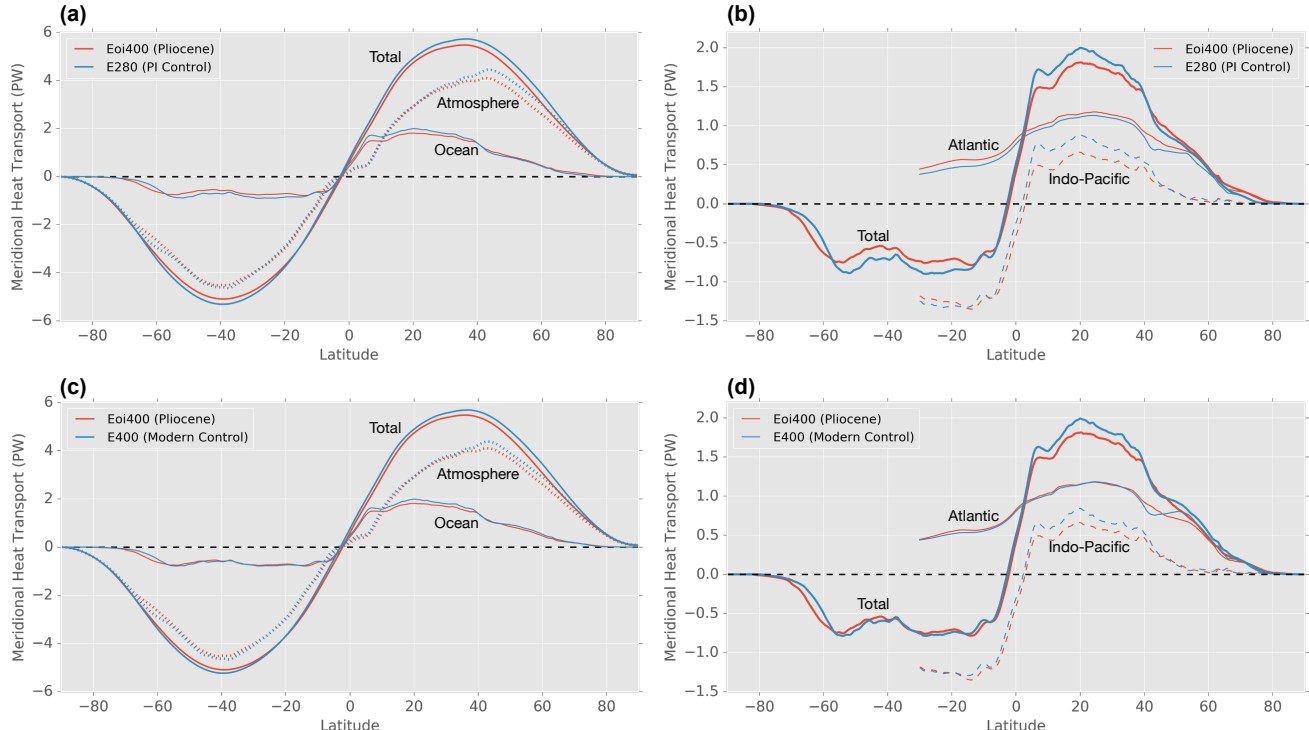

**Figure 15.** Total meridional heat transport and its decomposition. The top (bottom) row compares the $\mathrm{E}oi^{400}{}_\mathrm{P}$ mid-Pliocene simulation $\langle 2691, 2720 \rangle$ to the PI $\mathrm{E}^{280}{}_\mathrm{P}$ $\langle 5131, 5160 \rangle$ (modern $\mathrm{E}^{400}{}_\mathrm{P}$ $\langle 1931, 1960 \rangle$) control. The left column shows the total meridional heat transport and its decomposition into the atmospheric and oceanic components. The right column decomposes the oceanic components into the transport in the Atlantic and the Indo-Pacific basins. The $\langle \rangle$ denotes the range of model years over which the climatology was computed.

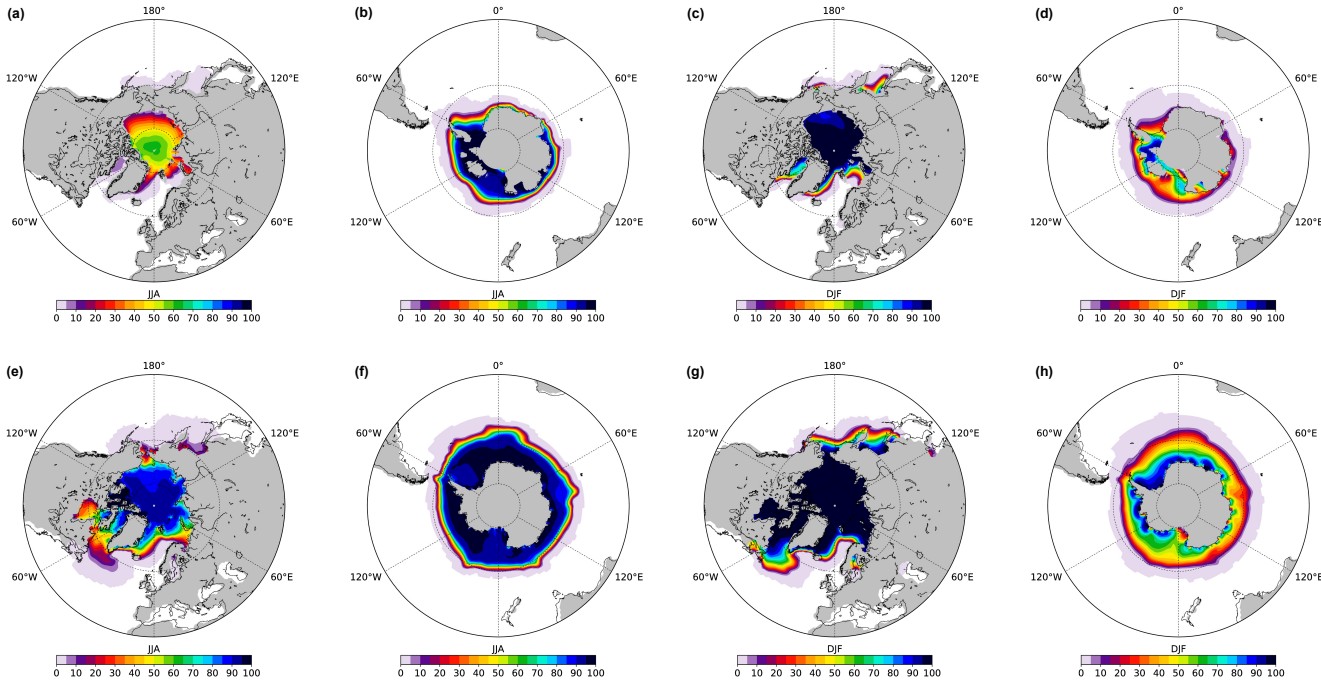

**Figure 16.** Surface area covered by sea ice for the mid-Pliocene ⟨2691, 2720⟩ (a)-(d) and PI control ⟨5131, 5160⟩ (e)-(h). The climatological mean for the austral winter months JJA is presented in the first two columns, and the boreal winter months DJF in the last two columns. The ⟨⟩ denotes the range of model years over which the climatology was computed.

**Table 1.** Configuration common to all experiments described in this paper.

| | |
|---|---|
| $CH_4$ | 760 ppb |
| $N_2O$ | 270 ppb |
| CFCs | 0 |
| $O_3$ | Local modern |
| Solar Constant | $1365 W/m^2$ |
| Eccentricity | 0.016724 |
| Obliquity | $23.446°$ |
| Perihelion | $102.04°$ |
| Dynamic vegetation | Off |
| Deep-water overflow parametrization | Off |

**Table 2.** Models details and diagnostics.

| Model | Description | $CO_2$ (ppmv) | Vertical mixing | Simulation Length (yr) | Energy Balance TOA ($Wm^{-2}$) | AMOC strength (Sv) | Climatology Years |
|---|---|---|---|---|---|---|---|
| $E^{280}$ | PI Control | 280 | $0.16\,cm^2s^{-1}$, constant | $4,650^a$ | 0.02 | 20 | $4,601 - 4,630$ |
| $E^{400}$ | Modern Control | 400 | $0.16\,cm^2s^{-1}$, constant | 1,900 | 0.08 | 21.9 | $1,871 - 1,900$ |
| $Eoi^{400}$ | Pliocene | 400 | $0.16\,cm^2s^{-1}$, constant | 1,350 | 0.06 | 21.2 | $1,221 - 1,250$ |
| $Eoi^{450}$ | Pliocene | 450 | $0.16\,cm^2s^{-1}$, constant | 1,150 | 0.14 | 21.4 | $1,121 - 1,150$ |
| $E^{280}{}_P$ | PI Control | 280 | POP1 type | $5,200^{ab}$ | 0.11 | 21.5 | $5,131 - 5,160$ |
| $E^{400}{}_P$ | Modern Control | 400 | POP1 type | $2,010^c$ | 0.17 | 24.2 | $1,931 - 1,960$ |
| $Eoi^{400}{}_P$ | Pliocene | 400 | POP1 type | $2,820^d$ | 0.1 | 23.4 | $2,691 - 2,720$ |
| $Eoi^{450}{}_P$ | Pliocene | 450 | POP1 type | $2,780^e$ | 0.1 | 23.7 | $2,621 - 2,650$ |

[a]Includes 3,500 years from existing control simulation
[b]Includes 630 years from model $E^{280}$
[c]Includes 360 years from model $E^{400}$
[d]Includes 1,120 years from model $Eoi^{400}$
[e]Includes 1,050 years from model $Eoi^{450}$

**Table 3.** Ocean temperature trends globally, and for the different oceanic depths considered in Figure 3. All values are in units of °C/century.

| Model | Global Ocean | SST | Upper Ocean | Middle Ocean | Lower Ocean |
|---|---|---|---|---|---|
| E$oi^{400}$P | 0.05 | 0.02 | 0.04 | 0.05 | 0.05 |
| E$oi^{450}$P | 0.05 | 0.04 | 0.03 | 0.05 | 0.06 |
| E$^{280}$P | 0.03 | 0.01 | 0.01 | 0.02 | 0.05 |
| E$^{400}$P | 0.06 | 0.01 | 0.03 | 0.05 | 0.07 |

**Table 4.** Proxy reconstructed mid-Pliocene Mean annual surface air temperatures (MASAT) during the PRISM3 interval (3.3 - 3.0 Mya) from *Salzmann et al.* (2013) compared to that simulated with our 400 ppmv mid-Pliocene simulation E$oi^{400}$P.

| Location | Continent | Latitude | Longitude | Proxy inferred SAT | Model SAT |
|---|---|---|---|---|---|
| Chara Basin, Siberia | Asia | 56.97 | 118.31 | 12.8 | -7.11 |
| Lake Baikal | Asia | 55.69 | 108.37 | $7.0 \pm 2.5$ | -0.90 |
| James Bay Lowland | North America | 52.83 | 276.12 | $6.0 \pm 2.0$ | 2.90 |
| Lower Rhine Basin | Europe | 51.03 | 6.53 | $14.1 \pm 0.2$ | 12.38 |
| Sessenheim-Auenheim | Europe | 48.82 | 8.01 | $14.6 \pm 0.7$ | 14.12 |
| Alpes-Maritimes | Europe | 43.82 | 7.19 | $17.5 \pm 2.0$ | 20.16 |
| Tarragona | Europe | 40.83 | 1.13 | $20.0 \pm 2.5$ | 20.19 |
| Rio Maior | Europe | 39.35 | 351.07 | $16.0 \pm 2.0$ | 20.05 |
| Yorktown, Virginia | North America | 36.59 | 283.62 | 17.5 | 19.34 |
| Andalucia G1 | Europe | 36.38 | 355.25 | $21.0 \pm 2.0$ | 21.90 |
| Habibas | Africa | 35.73 | 358.88 | $21.0 \pm 1.0$ | 22.10 |
| Nador | Africa | 35.18 | 357.07 | $21.5 \pm 1.0$ | 19.40 |
| Pinecrest, Florida | North America | 27.36 | 277.56 | 23.1 | 26.10 |
| Hadar | Africa | 11.29 | 40.63 | $20.5 \pm 1.0$ | 20.86 |

**Table 5.** Mean annual surface air temperatures (MASAT) and Earth System Sensitivity (ESS)

| Model | MASAT (°C) | $\Delta T$ with E$^{280}$P (°C) | $\Delta T$ with E$^{400}$P (°C) | ESS (°K/$2 \times CO_2$) |
|---|---|---|---|---|
| E$oi^{400}$P | 16.8 | 3.8 | 1.8 | 7.4 |
| E$oi^{450}$P | 17.3 | 4.3 | 2.3 | 6.3 |
| E$^{280}$P | 13.0 | — | — | — |
| E$^{400}$P | 15.0 | — | — | — |

**Table 6.** Mean seasonal surface air temperatures and anomalies.

| Season | Control | | Pliocene | | Anom w.r.t $E^{280}{}_P$ | | Anom w.r.t $E^{400}{}_P$ | |
|--------|---------|---------|---------|---------|---------|---------|---------|---------|
| | $E^{280}{}_P$ | $E^{400}{}_P$ | $Eoi^{400}{}_P$ | $Eoi^{450}{}_P$ | $\Delta Eoi^{400}{}_P$ | $\Delta Eoi^{450}{}_P$ | $\Delta Eoi^{400}{}_P$ | $\Delta Eoi^{450}{}_P$ |
| DJF | 11.3 | 13.3 | 14.4 | 15.0 | 3.1 | 3.7 | 1.1 | 1.7 |
| MAM | 12.9 | 14.9 | 16.7 | 17.3 | 3.8 | 4.4 | 1.8 | 2.4 |
| JJA | 14.9 | 16.8 | 19.2 | 19.7 | 4.3 | 4.8 | 2.4 | 2.9 |
| SON | 13.0 | 15.0 | 16.7 | 17.2 | 3.7 | 4.2 | 1.7 | 2.2 |