# Peer review of "Regional and global climate for the mid-Pliocene using the University of Toronto version of CCSM4 and PlioMIP2 boundary conditions"

_Climate of the Past, 2017_

## Referee Comment (RC1) · Anonymous Referee #1 · 3 Apr 2017

Mr. Deepak Chandan and Dr. Richard Peltier (DR for short) presented a new set of PlioMIP2 simulations and comparisons with PI simulation in order to understand the large-scale mid-Piacenzian climate features (mPWP). The writing is clear, and the introduction is very well written. However, considering the large differences in the Pliocene temperature responses and the climate sensitivity derived from the results of CCSM4 reported in IPCC AR5, and Rosenbloom et al., 2013, which also uses CCSM4, a number of critical issues need to be addressed before this paper is publishable.

The results presented here exhibit appreciable differences from previously published PI and PlioMIP1 experiments with CCSM4. The claim that this is due to the extended simulation length is not convincing as the surface temperature field appears to be

in quasi-equilibrium after about 500 years (Fig. 5); extending the simulation beyond this point only adds about 0.5°C to the global annual temperature. The authors are urged to (1) double-check their simulation setup, (2) run a PlioMIP1 simulation using the same setup as in Rosenbloom et al., (2013) to assess differences from previously published experiments with CCSM4 (data can be downloaded from the Earth System Grid: https://www.earthsystemgrid.org/dataset/ucar.cgd.ccsm4.output.html). DR's simulations (if proven valid) may be interesting for the purpose of testing CCSM4 parameterizations, but they may not be appropriate for PlioMIP2. One major goal of PlioMIP2 is to evaluate skills in simulating mPWP climate by state-of-the-art Earth-system models. DR's PI simulation shows much worse model skills (too cold and too much sea ice) compared to the published benchmark CCSM4 PI simulation. It is confusing to use these simulations to serve the intercomparison purpose for CCSM4. Finally, there is a lack of information on the spin-up and diagnostics of DR's mPWP simulations, which hinders comprehensive evaluations of validity of these simulations. I am highlighting major differences from published CCSM PI runs here, using the part of DR simulations that do not include changes to ocean diapycnal diffusivity: 1. The global surface temperatures in DR's PI simulation is between 12.5 to 13 °C (Fig. 5 and Table 2), which is 1°C colder than CCSM4 PI benchmark simulation (http://www.cesm.ucar.edu/experiments/cesm1.0/diagnostics/b40.1850.track1.1deg.006/atm_863-892-obs/); the benchmark simulation shows 13 °C for 2-m air temperature and 14 °C for surface temperature. Northern hemispheric sea ice in DR's PI case covers Labrador Sea and North Pacific at least seasonally (Fig. 15), which is way too extensive compared to the published PI benchmark simulation (Gent et al., 2011). In Gent et al., (2011), the CCSM4 PI benchmark has an AMOC strength > 24 Sv. Shown in Yeager and Danabasoglu (2012) Figure 1b, turning off the overflow parameterization does not change the AMOC strength much, which is measured as the maximum overturning of the North Atlantic. However, in DR's PI simulation, the AMOC strength is only 20 Sv (Table 4.). Finally, the entire ocean in DR's PI simulation is colder than what is reported by Gent et al (2011). 2. The climate sensitivity to a doubling of

CO2 is ~4°C (based on simulations E400 and E280), while the estimate for CCSM4 is 3.2°C/per doubling of CO2 (Bitz et al., 2011). 3. Comparing the zonal mean temperatures in DR's E400 to E280 case (the distance between red and green lines in Fig. 8), the 120 ppm CO2 increase leads to ~7°C warming in Arctic region. In IPCC AR5 (Fig 12.11 in Collins et al., 2013), ~100 ppm CO2 increase between 2081-2100 and 1986-2005 based on RCP2.6 simulations shows <3-4°C Arctic warming.

More information is needed to evaluate DR's mPWP simulations: 1. Details about the pre-existing 3500 year spin up simulation should be presented. This run was used to initialize the E280 and E400 case. For example, what are the salinity trends in major ocean basins and what are the sea ice trends? how were the carbon and nitrogen pools initialized and spun up to equilibrium? Moreover, how does the simulated E400 climate state compare to present-day observations? The surface temperatures presented in Fig. 5 suggest that both E280 and E400 start from a very cold ocean state, which is an odd practice for simulating PI or present-day climate. 2. Why are the Pliocene simulations initialized from Levitus? This shows a very different provenance from the controls. Earth system sensitivity analyses should not be performed when the oceans are started so differently and not brought into any kind of equilibrium (clear ocean temperature trends are shown in Figure 4 for PlioMIP2 cases). Based on Fig. 4, difference in ocean temperature between E400 and Eoi400 could be an artificial result of different initialization. 3. The argument for vegetation mapping is weak (P8, Line 30-34). For example, for Megabiome 1, 80% PFT type 5 and 20% PFT type 7 could lead to different surface albedo from 20% PFT type 5 and 80% PFT type 7. More rigorous mapping method should be derived and applied to the PlioMIP2 simulations. 4. The significant and widespread cooling in DJF over Canada and Eurasia in the Eoi400 simulation as compared to the E400 simulation needs to be explained (Figure 7). The cooling suggests a possible problem with the initialization of the land surface in the Eoi400 simulation. Please show time series of net primary productivity and leaf area index. An example of Rosenbloom et al (2013) can be found here: http://webext.cgd.ucar.edu/B1850/Pliocene/lmwg/ccsm2/b40.plio.FV1.003extb40.1850.track1.1deg.006/set1/set1.html. 5. Global, annual surface air temperature in the Eoi400 simulation is ∼2C warmer than in the E400 simulation (Figure 5, Table 2). This is puzzling, even more so because ∼1C of this surface temperature difference occurs at tropical latitudes (Figure 8) where the relatively small coverage and changes in vegetation would not be expected to produce that much warming. If this is indeed a result of the changes in the gateways then a figure supporting this contention needs to be included. 6. As noted above, DR's PI simulation shows too much sea ice in the North Atlantic and Labrador Sea compared to the published benchmark CCSM4 PI simulation. As such, it strongly influences their surface air temperature anomalies in the North Atlantic and Labrador Sea, as shown in Figures 6a and 7a, leading to their conclusion that they better match the PRISM3 SST anomalies in these regions (Figure 12) than the simulation of Rosenbloom et al. In fact, their PlioMIP2 simulation, by itself, is colder than Rosenbloom et al., (2013).

Detailed comments:

P 1, Line 9-10: This is a different way in which the CCSM4 has been used with major changes to the ocean vertical mixing physics, so not just boundary conditions are different. P 3, Lines 22-25: Cite Haywood et al (2011) when talking about the designs and aim of PlioMIP1 P 6, lines 10-15: Please discuss the choice of the ocean convective parameterization. Was the KPP scheme used apart from turning off the tidal mixing component? Was tidal mixing also turned off when the Kv was kept constant? Perhaps show your preindustrial POP1 simulation's mixed layer depths. How is the 2-d distribution of mixed layer depth different from the standard CCSM4 preindustrial simulation as described in Danabasoglu et al., J. Climate, 25, 2012? P 7, line 8: The default 1° POP2 bathymetry field (KMT) is not from GTOPO30 as described here. The CCSM4 POP2 KMT field was generated from ETOPO2v2 as described and cited by Danabasoglu et al. 2012. P 8, 18-20: This line, describing how the global mean salinity is adjusted for glacial climates to reflect a drop in sea level, and hence ocean volume, is not needed here as there is no glacial climate simulation. Thus, to make the manuscript more con-

Interactive
comment

cise and focused, it should be deleted. P9, line 22: By what measure have the global mean SSTs come into equilibrium? Trends are still quite apparent in Fig. 3a. P9, line 24: "...increases the rate of warming for the middle and lower and decreases the rate of warming for the upper ocean." This decrease/increase of trend is not apparent apart from the very beginning of the start of the POP1-Kv runs. The trends at the end of the POP1-Kv runs look very similar to the trends at the end of the POP2-Kv runs. Typically there is a flattening as the ocean model equilibrates, but this is not the case here. P9, line 29: "...given the deep ocean sufficient time to come into equilibrium." Again, Fig 3d shows that the deep ocean is far from equilibrium with major drifts by the end of the simulations. Comparing Figs 3b-d, there does seem to be a transfer of heat from the upper ocean at depths above 550 m which leads to cooling at the beginning of the POP1-Kv runs, while the deep ocean below 1850m starts to warm. There was probably not a net heat loss to the atmosphere at the start through venting the upper ocean, otherwise the global mean temperature (Fig. 4) would show a cooling trend at this time instead of a warming trend. P9, lines 30-32: Trends of global mean temperatures are twice as large O(.07K/century, shown in Fig. 2) as what is shown in Rosenbloom et al. 2013 for the CCSM4 PlioMIP1 simulation, which is around 0.03K/century. P13, lines 12-13: "...run to near statistical equilibrium..." By what measure are these simulations in a statistical equilibrium? As discussed above, deep ocean temperature trends are large and the global mean ocean temperature at the end of the runs exceeds the CCSM4 PlioMIP1 simulation discussed in Rosenbloom et al. 2013. P13, lines 18-34: Please indicate over what region of the water column and latitude range the maximum Atlantic Meridional Overturning Circulation metric was calculated. One needs to exclude the upper 500m or so of the wind-driven tropical cells. Figure 5 and Table 2 temperature both say Mean Annual Surface Air Temperature, but in comparing the same runs, the values do not agree. Table 2 numbers are higher than shown in Figure 5, suggesting that Table 2 numbers are TS, the Surface Temperature (radiative) which tends to be higher than the TREFHT or 2m air temperature typically used for 'MASAT'. Figure 10: It is difficult to discern the contours in 10b-c. There appears to be possibly a very large

cold bias in c and a smaller one in b, based on the minima shown in the color bars for these two panels of anomalies. There appears to be a large black swath in panel c emanating from the eastern boundary of in the N. Pacific, suggesting a very large cold anomaly with respect to the Modern control.

References Collins, M., et al. (2013), Chapter 12: Long-term climate change: Projections, commitments and irreversibility, in Working Group 1 Contribution to the IPCC Fifth Assessment Report—Climate Change: The Physical Science Basis. Danabasoglu, G., Yeager, S.G., Kwon, Y.O., Tribbia, J.J., Phillips, A.S. and Hurrell, J.W., 2012. Variability of the Atlantic meridional overturning circulation in CCSM4. Journal of Climate, 25(15), pp.5153-5172. Gent, P.R., Danabasoglu, G., Donner, L.J., Holland, M.M., Hunke, E.C., Jayne, S.R., Lawrence, D.M., Neale, R.B., Rasch, P.J., Vertenstein, M. and Worley, P.H., 2011. The community climate system model version 4. Journal of Climate, 24(19), pp.4973-4991. Haywood, A. M., H. J. Dowsett, M. M. Robinson, D. K. Stoll, A. M. Dolan, D. J. Lunt, B. Otto-Bliesner, and M. A. Chandler (2011), Pliocene Model Intercomparison Project (PlioMIP): Experimental design and boundary conditions (Experiment 2), Geosci. Model Dev., 4(3), 571–577,doi:10.5194/gmd-4-571-2011 Rosenbloom, N.A., Otto-Bliesner, B.L., Brady, E.C. and Lawrence, P.J., 2013. Simulating the mid-Pliocene Warm Period with the CCSM4 model. Geoscientific Model Development, 6(2), pp.549-561. Yeager, S. and Danabasoglu, G., 2012. Sensitivity of Atlantic meridional overturning circulation variability to parameterized Nordic Sea overflows in CCSM4. Journal of Climate, 25(6), pp.2077-2103.

---

## Referee Comment (RC2) · Anonymous Referee #2 · 6 Apr 2017

In this paper, based on the PlioMIP2 boundary conditions, the authors use the CCSM4 to simulate the mid-Pliocene warm climate. The experiments are well designed and described in the paper. The paper is well written, and provide very good summaries for the Pliocene studies from aspects of ocean, land-ice reconstructions.

However, in the paper, the authors do not clarify why they choose different diapycanl mixing k to carry out their experiments. In their models outputs, it is clear that the depth-dependent k remarkable changes the simulated ocean climate. Why do the authors only compare the simulated Pliocene and PI climate in the experiments with depth-dependent k? Do these experiments provide even stronger warming in the high-latitudes? In the revised version, I suggest the authors to plot figures similar to Figure

6, but based on the experiments with the constant k.

I also suggest the authors to plot AMOC for both two sets of experiments. It is helpful for other groups to judge which k provides better simulations for AMOC.

Since this work is taking part in the PlioMIP2, the authors should provide suggestions for other groups which set of experiments should be used in the future intercomparsions, the set with the constant k or the set with the depth-dependent k. Why?

———————————————

---

## Author Comment (AC1) · 24 Apr 2017

**Response to Referee 1**

We are very grateful to the referee for the time and effort expended in reviewing our manuscript. We believe that these comments and suggestions have enabled us to improve the presentation insofar as its clarity is concerned.

> The results presented here exhibit appreciable differences from previously published PI and PlioMIP1 experiments with CCSM4. The claim that this is due to the extended simulation length is not convincing as the surface temperature field appears to be in quasi-equilibrium after about 500 years (Fig. 5); extending the simulation beyond this point only adds about 0.5°C to the global annual temperature.

It is important to note that we have never claimed in our manuscript that our ability to capture PRISM3 SST anomalies much more accurately with our models than was possible with CCSM4 in the first phase of PlioMIP is only (or even primarily) due to the very long timescale over which the simulations have extended. In fact, we have made it clear that it is the nature of the new boundary conditions that have contributed primarily to the much improved data-model comparison that our work has achieved. However, the implementation of these new boundary conditions has required significantly longer integration times to reach conditions that are sufficiently close to equilibrium, as we argue in greater detail below in response to another of the reviewer's comments. Note specifically the following excerpts from the submitted text:

Abstract: "With the new boundary conditions, the CCSM4 model simulates a mid-Pliocene which is..."

Conclusion: "We find that the PRISM4 boundary conditions mandated in PlioMIP2 lead to greater warming in the mid-Pliocene"

It is also incorrect to assume that because the globally averaged SST might have reached quasi-equilibrium that this also implies that the SSTs have come to equilibrium regionally. Since, at year 500, the MOC continues to move towards its new equilibrium and the TOA energy imbalance continues to decrease, the model is still in a state of significant disequilibrium.

> The authors are urged to (1) double-check their simulation setup, (2) run a PlioMIP1 simulation using the same setup as in Rosenbloom et al., (2013) to assess differences from previously published experiments with CCSM4 (data can be downloaded from the Earth System Grid: https://www.earthsystemgrid.org/dataset/ucar.cgd.ccsm4.output.html).

Given the significant difficulties insofar as the utility of model-data comparisons based upon the initial design of the PlioMIP effort is concerned, it is our opinion that nothing could possibly be gained by regressing to that initial design and wasting the very significant computational resources that this would entail. This is especially the case given the very marked improvement of the data-model inter-comparisons that have been achieved with the new experimental design. Our analyses are the one of the first to be produced based upon this improved design.

> DR's simulations (if proven valid) may be interesting for the purpose of testing CCSM4 parameterizations, but they may not be appropriate for PlioMIP2. One major goal of PlioMIP2 is to evaluate skills in simulating mPWP climate by state-of-the-art Earth-system models. DR's PI simulation shows much worse model skills (too cold and too much sea ice) compared to the published benchmark CCSM4 PI simulation. It is confusing to use these simulations to serve the intercomparison purpose for CCSM4.

These comments are also ill-founded for reasons that will be further clarified in what follows. In fact, the pre-industrial control model employed in *Rosenbloom et al.* [2013], while it provided a reasonable representation of the climate of that era, suffered from significant flaws as a control model on the basis of which one might attempt to ascertain the deviation of mPWP climate from that of the pre-industrial era. In particular, the PI control employed by *Rosenbloom et al.* had both the "overflow" parameterization and the "tidal mixing" parameterizations turned on. Given that the new boundary conditions have significantly modified bathymetric depth throughout the global oceans and given that the land sea mask is also modified, it is expected that these modern climate-based parameterizations would be inappropriate for the mPWP on a priori grounds. We have therefor turned them off to produce a unique PI control with which our mPWP results can be compared, unencumbered by the inconsistency of the comparisons previously produced using CCSM4 in the context of the original PlioMIP exercise. This is further discussed in what follows.

> Finally, there is a lack of information on the spin-up and diagnostics of DR's mPWP simulations, which hinders comprehensive evaluations of validity of these simulations. I am highlighting major differences from published CCSM PI runs here, using the part of DR simulations that do not include changes to ocean diapycnal diffusivity:

Before we individually address the specific differences between our CCSM4 PI and *Gent et al.* [2011] PI that the reviewer has referred to, it is necessary to make it clear that *Gent et al.* is but a single realization of the PI control run which was produced by the authors for a single specific configuration of the CCSM4 model (overflow and tidal mixing parameterizations turned on) and for a specific number of model years. It is entirely expected that different configurations of the same model, and different lengths of model integration will result in a set of climatologies that differ somewhat among them, while being broadly similar. There is no single PI simulation that can be proposed to be better than any other PI simulation because such a determination cannot be made for a time long before the start of modern observational era. What is actually important is that the configuration of the model employed for the PI control not contain parameterizations that are expected not to be applicable under mPWP conditions.

Here are some of the ways in which our PI simulation differs from *Gent et al.* [2011].

1. The *Gent et al.* simulation was integrated for a total of only 1,300 model years whereas our PI control run has been integrated for a total of 5,170 years which is almost four times longer than the "benchmark simulation". There is no physical basis on which the climatology of two simulations differing by almost 4,000 years in model years should agree.

2. We share our own concern with that of the referee in regards to the importance of establishing a steady state control with minimal model drift (discussed further below), and it is for that reason that we have undertaken the very challenging task of simulating a 1° coupled atmosphere-ocean PI control for so many model years. This has resulted in a model that is much closer to equilibrium than the control of *Gent et al.*, which continued to lose heat at the top of the atmosphere (TOA) at a globally averaged rate of $-0.147\,W\,m^{-2}$. This is larger than the $< |0.1|\,W\,m^{-2}$ energy loss rate that *Gent et al.*, themselves consider desirable and larger than the $0.11\,W\,m^{-2}$ energy balance that we get in our PI control $E^{280}{}_P$. Furthermore, the $E^{280}$ model from which $E^{280}{}_P$ was branched off reached an energy balance of $0.02\,W\,m^{-2}$ implying an essentially perfect equilibrium state. Continued integration of $E^{280}{}_P$ will lead to further reduction in the TOA energy imbalance.

3. The ocean model in *Gent et al.* CCSM4 PI simulation was configured with both (i) the overflow parametrization and (ii) tidal mixing scheme turned on. As we have discussed in our manuscript, and which we further discuss below in response to the reviewer's concerns, these two options of the ocean model are tuned to the present day conditions and as such should not be used in palaeo climate simulations [see *Vettoretti and Peltier* [2013], *Peltier and Vettoretti* [2014] for comments on this point in connection with their work on the Dansgaard-Oeschger oscillation phenomenon] for epochs in which they are expected to be inappropriate on a priori grounds.

4. There are also some differences between the ocean mask in our PI control and that in *Gent et al.* See *Vettoretti and Peltier* [2013] for details of the differences. Of course the land-sea mask under mPWP conditions differs significantly from modern and this will have a strong impact upon the tidal regime on the basis of which a new tidal mixing parameterization, appropriate to that era, would have to be determined.
* * *
1. The global surface temperatures in DR's PI simulation is between 12.5 to 13°C (Fig. 5 and Table 2), which is 1°C colder than CCSM4 PI benchmark simulation (`http://www.cesm.ucar.edu/experiments/cesm1.0/diagnostics/b40.1850.track1.1deg.006/atm_863-892-obs/`); the benchmark simulation shows 13°C for 2-m air temperature and 14°C for surface temperature. Northern hemispheric sea ice in DR's PI case covers Labrador Sea and North Pacific at least seasonally (Fig. 15), which is way too extensive compared to the published PI benchmark simulation (Gent et al., 2011). In Gent et al., (2011), the CCSM4 PI benchmark has an AMOC strength > 24 Sv. Shown in Yeager and Danabasoglu (2012) Figure 1b, turning off the overflow parameterization does not change the AMOC strength much, which is measured as the maximum overturning of the North Atlantic. However, in DR's PI simulation, the AMOC strength is only 20 Sv (Table 4.). Finally, the entire ocean in DR's PI simulation is colder than what is reported by Gent et al (2011).
* * *
The minor differences that the reviewer has noted between our PI control and that by *Gent et al.* [2011] are entirely expected on the physical grounds that our PI control has been integrated for a period that is four times longer than that in *Gent et al.* and has both the overflow

parameterization and tidal mixing parameterization turned off. Therefore, the reported climatological features of our model represent a different (and more fully developed) equilibrium state than that in *Gent et al.*

The temperature reported in Figure 5 of the manuscript is not 2 m air temperature, which is something we clarify below in response to another of the referee's comments. This has also been clarified in the revised manuscript. The values in Table 2 are the 2 m air temperature which for our PI control model is 13°C, a value which is in very good agreement with the 2 m air temperature for the benchmark simulation (13°C according to the referee).

With regards to the sea ice in the Labrador Sea and the North Pacific, these differences are due to the different configurations of the ocean component, and the significantly different duration of the simulations, as discussed above. Furthermore, in the absence of any reliable observations regarding the extent and volume of PI sea ice there is no reason to assume that either the result of our simulation or that of *Gent et al.* is superior to the other. However, it does appear questionable as to why the margins of the mean annual arctic sea ice concentration in the *Gent et al.* PI control are as close to that which has been inferred on the basis of modern satellite observations (Figure 3 in *Gent et al.*, also reproduced below)? The equilibrium state of the 280 ppmv PI climate would be expected, on *a priori* grounds, to be characterized by more extensive Arctic sea ice formation than that which has been observed for the recent decades and this is what is predicted by our unique PI control model. The following Figure compares the mean annual arctic sea ice in our PI (left) and in *Gent et al.* (right). The sea ice coverage in our PI is only slightly larger ($\sim 16\%$) than *Gent et al.* ($1.36 \times 10^7 \, km^2$ in ours compared with $\sim 1.17 \times 10^7 \, km^2$ in *Gent et al.*).

[Figure]

Interestingly, the PI control that has been used by *Rosenbloom et al.* [2013] in PlioMIP (which is an extension of *Gent et al.*, in that the reported PI climatology is from the last 100 years of the 1,300 year *Gent et al.* control, whereas *Gent et al.* report on years 871-900 of that same simulation) has more arctic sea ice than our PI control. The Figure on the next page shows the monthly cycle of Arctic sea ice extent for our PI control (left), and the PI controls of all PlioMIP

models on right (from *Howell et al.* [2016]). We note two things (i) the large spread among the PlioMIP models for the PI sea ice seasonal cycle, particularly during the winter months, and (ii) our PI seasonal cycle follows the ensemble mean closely, especially during June-December. Annually averaged, the sea ice extent in our CCSM4 PI is $1.36 \times 10^7 \, km^2$ and that in *Rosenbloom et al.* [2013] is $1.83 \times 10^7 \, km^2$ (as reported in *Howell et al.* [2016]), further arguing against the referee's assumption that there is there is too much sea ice in our PI control model. These comparisons leave no doubt as to the reasonableness of our PI sea ice cover.

[Figure]

The AMOC strength that the reviewer is quoting is from the version of our PI control with the pelagic value of the diapycnal diffusivity. Our PI control with the more realistic POP1 profile of diapycnal diffusivity that is employed in the KPP parameterization is 21.5 Sv. Observational estimates for the strength of the AMOC are in the range of 17-18 Sv [*Cunningham et al.*, 2007, *McCarthy et al.*, 2015]. Therefore the 24 Sv estimate by *Gent et al.* [2011] is the problem, and our own estimates are larger than the observationally estimated range, although, compared with *Gent et al.* our results are closer to observations. The CCSM4 PI AMOC strength that was used for the intercomparison of AMOC among PlioMIP models was even larger at 25.7 Sv [*Zhang et al.*, 2013]. It seems that CCSM4 has a consistent bias towards predicting AMOC strengths that are too large. In the intercomparison by [*Zhang et al.*, 2013] the CCSM4 AMOC strength was an outlier among the eight models compared; five of the models predicted an AMOC in the range of 14-19 Sv.

With regards to the reviewer's comment that elimination of the overflow parametrization does not change the strength of the AMOC "much", we show on the figure on the next page, a preliminary result that turning off the overflow parametrization leads to a change of $\sim 1$ Sv (comparing green and red curves). The red curve is the AMOC strength as predicted by the NCAR configuration for PI (i.e. that of *Gent et al.*), the blue curve is the same NCAR model which has tidal mixing and overflow parametrization turned off. Finally, the black curve shows the PI control from which our control simulations $E^{280}$ and $E^{400}$ are started. This speaks to the excellent equilibrium of the pre-existing control from which our PlioMIP controls are initiated.

The referee's claims that the entire ocean in our PI is colder than *Gent et al.* is manifestly incorrect. The globally averaged ocean temperature reported for the PI in *Gent et al.* is 3.13°C, whereas the globally averaged temperature in our PI ($E^{280}$P) towards the end of the simulation is $3.1 - 3.2$°C.
* * *
2. The climate sensitivity to a doubling of CO2 is $\sim 4$°C (based on simulations E400 and E280), while the estimate for CCSM4 is $\sim 4$°C/per doubling of CO2 (Bitz et al., 2011).
* * *
[Figure]

The reviewer is confusing the Equilibrium Climate Sensitivity (ECS) which is the subject of *Bitz et al.* [2012] with Earth System Sensitivity (ESS), which is the subject of our paper and which has been employed in the context of PlioMIP [*Haywood et al.*, 2013]. ECS (also sometimes known as the Charney Sensitivity) is defined as the change in the average surface temperature of the planet in response to a doubling of $CO_2$ concentrations, determined after a sufficient amount of time has passed to allow for the fast feedback components of the climate system such as water-vapor, aerosols, clouds, sea-ice albedo etc. to reach equilibrium. In contrast, a model's ESS is computed after allowing sufficient time for the slow feedback processes such as those associated with ocean circulation, and land ice and vegetation (the latter two only applicable if the climate model is configured with them) to reach equilibrium. Consequently, ESS is always larger than ECS. In palaeo-climate simulations where one's interest is in the equilibrium state it is the ESS that is of interest. For the original PlioMIP, the CCSM4 model's ESS was determined to be $3.51°C/2 \times CO_2$ [*Haywood et al.*, 2013] while the ECS of CCSM4 is $3.2°C/2 \times CO_2$ [*Bitz et al.*, 2012].

3. Comparing the zonal mean temperatures in DR's E400 to E280 case (the distance between red and green lines in Fig. 8), the 120 ppm $CO_2$ increase leads to $\sim 7°C$ warming in Arctic region. In IPCC AR5 (Fig 12.11 in Collins et al., 2013), $\sim 100$ ppm $CO_2$ increase between 2081-2100 and 1986-2005 based on RCP2.6 simulations shows $< 3 - 4°C$ Arctic warming.

The referee has here made the further error of confusing the temperature anomaly between two equilibrium climates with the temperature anomaly between two transient states. This error is related to the reviewer's mistake in confusing ESS and ECS which we have discussed above. The climates for years 2081-2100 and for years 1986-2005 are not in equilibrium, rather the climate in these IPCC models is continually forced by historical and RCP emission scenarios. In each of these two intervals the slow feedback components of the climate system haven't even come close to equilibrium, and one might even question as to how accurate it is to assume that the fast feedback processes might have reached equilibrium while the climate system continues

to be forced by time varying emissions. Therefore, the arctic warming in *Collins et al.,* [2013] is expected to be smaller than that which would be obtained when comparing two multi-millennia climate runs such as $E^{280}$ and $E^{400}$.

> More information is needed to evaluate DR's mPWP simulations: 1. Details about the pre-existing 3500 year spin up simulation should be presented. This run was used to initialize the E280 and E400 case. For example, what are the salinity trends in major ocean basins and what are the sea ice trends? how were the carbon and nitrogen pools initialized and spun up to equilibrium? Moreover, how does the simulated E400 climate state compare to present-day observations? The surface temperatures presented in Fig. 5 suggest that both E280 and E400 start from a very cold ocean state, which is an odd practice for simulating PI or present-day climate.

We refer the reviewer to *Vettoretti and Peltier* [2013] for a discussion of the pre-existing 3,500 year spin up. The ocean salinity and temperature trends and the sea-ice trend are shown below for the pre-existing spin-up. These analyses show that the spin-up simulation is in a steady-state equilibrium. The global ocean temperature trend over the last 500 years is only $0.01°C/century$.

[Figure]

A comparison between the simulated $E^{400}$ climate state and present-day observations is not part of the manuscript because:

1. Such a comparison is not pertinent to the topic of the mid-Pliocene.

2. The $E^{400}$ simulation in PlioMIP2 is not designed to represent the present-day. It does not include present day aerosols, CFCs, emissions scenario, urban land units and agricultural land units. Its sole purpose is to serve as a 400 ppmv control.

3. The $E^{400}$ leads to an equilibrium climate and therefore there isn't much value in comparing it to the present day transient climate. The CMIP project is devoted to understanding model inter-comparisons for the present-day.

The $E^{280}$ and $E^{400}$ are branched from the pre-existing control run, as such the temperature evolution of the two models begins near the equilibrated ocean state of the 3,500 year long PI control. It is not clear to us why the reviewer was confused by this given that the reviewer seems to understand that the two simulations are branched from a well equilibrated PI run (which was started from *Levitus and Boyer* [1994], which we clarify in the next point). Nevertheless, we hope that our response has clarified this for the reviewer.

2. Why are the Pliocene simulations initialized from Levitus? This shows a very different provenance from the controls. Earth system sensitivity analyses should not be performed when the oceans are started so differently and not brought into any kind of equilibrium (clear ocean temperature trends are shown in Figure 4 for PlioMIP2 cases). Based on Fig. 4, difference in ocean temperature between $E^{400}$ and $Eoi^{400}$ could be an artificial result of different initialization.

The perpetually running control from which $E^{280}$ and $E^{400}$ control experiments were branched was also started from *Levitus and Boyer* [1994] [see *Vettoretti and Peltier* [2013]]. We realize that this information wasn't clear from our discussion of the control experiments in section 3.2 and we have made the changes in the revised version of the text needed to clarify this point. Therefore, all simulations we report upon have been initialized from *Levitus and Boyer* [1994].

3. The argument for vegetation mapping is weak (P8, Line 30-34). For example, for Megabiome 1, 80% PFT type 5 and 20% PFT type 7 could lead to different surface albedo from 20% PFT type 5 and 80% PFT type 7. More rigorous mapping method should be derived and applied to the PlioMIP2 simulations.

Paleo-vegetation reconstruction is a significant source of uncertainty that continues to plague all palaeo climate simulations including the simulations that have been published in the context of PlioMIP1 and those that are currently being carried out for PlioMIP2. In order to maintain consistency with PlioMIP1 simulations and the PlioMIP2 simulation using the MRI-CGCM2.3 model [*Kamae et al.*, 2016] we use the mid-Pliocene reconstruction by *Salzmann et al.* [2008]. The mapping from the mega-biomes to PFTs that we have employed is based on the relative projection of each of the mega-biomes onto PFTs for the modern day. Therefore, while there is a certain amount of latitude in the relative makeup of a mega biome in terms of PFTs, the example that the reviewer has used is unlikely because that situation would not constitute a reasonable projection of that mega biome onto PFTs. *Rosenbloom et al.* [2013] have used a similar methodology for initializing their vegetation cover, and although their choice of map was not made available as part of their model publication, we feel it is important for us to make our map available so that other groups might use it in their Pliocene simulations if they so choose.

4. The significant and widespread cooling in DJF over Canada and Eurasia in the Eoi400 simulation as compared to the E400 simulation needs to be explained (Figure 7). The cooling suggests a possible problem with the initialization of the land surface in the Eoi400 simulation. Please show time series of net primary productivity and leaf area index. An example of Rosenbloom et al (2013) can be found here: `http://webext.cgd.ucar.edu/B1850/Pliocene/lmwg/ccsm2/b40.plio.FV1.003ext-b40.1850.track1.1deg.006/set1/set1.html`.

We have mentioned in our manuscript that the nature of cooling in DJF over Canada and Eurasia in the anomaly of the Pliocene simulation with 400 ppmv modern bears special consideration due to its potential similarity to observed cooling trends over recent decades [*Cohen et al.*, 2013,

*Overland et al.*, 2011, *Sun et al.*, 2016] and therefore a comprehensive examination is planned to be the subject of a further publication.

As per the request of the reviewer, we show in the next figure the time series of NPP and TLAI over the last hundred years of simulation for both the 400 ppmv Pliocene and the control. These results show a robust and well equilibrated vegetation and there is no sign of any problem with the initialization of the land surface in the Pliocene simulation.

[Figure]

5. Global, annual surface air temperature in the Eoi400 simulation is $\sim 2°$C warmer than in the E400 simulation (Figure 5, Table 2). This is puzzling, even more so because $\sim 1°$C of this surface temperature difference occurs at tropical latitudes (Figure 8) where the relatively small coverage and changes in vegetation would not be expected to produce that much warming. If this is indeed a result of the changes in the gateways then a figure supporting this contention needs to be included.

The reviewer asserts that the simulated anomalies are "puzzling" but has not articulated as to why? With regards to the temperature difference along the tropical latitudes, this is due to the significant expansion of warm pool waters throughout the tropics globally (manuscript Figures 10, 11), with which the atmosphere exchanges latent and sensible heat.

6. As noted above, DR's PI simulation shows too much sea ice in the North Atlantic and Labrador Sea compared to the published benchmark CCSM4 PI simulation. As such, it strongly influences their surface air temperature anomalies in the North Atlantic and Labrador Sea, as shown in Figures 6a and 7a, leading to their conclusion that they better match the PRISM3 SST anomalies in these regions (Figure 12) than the simulation of Rosenbloom et al. In fact, their PlioMIP2 simulation, by itself, is colder than Rosenbloom et al., (2013).

The reviewer's comment that our simulation is colder than *Rosenbloom et al.* [2013] is puzzling considering that the volume-integrated global ocean temperature in our 400 ppmv Pliocene simulation is $\sim 5.8°$C (manuscript figure 4), compared to $\sim 4°$C in *Rosenbloom et al.* [2013] (Figure 2c). We point out that this is the second time that the reviewer has made

an incorrect statement regarding the comparison of ocean temperature between one of our simulations and one of the reviewer's preferred set of simulations. The first mistake was when the reviewer incorrectly claimed that our PI ocean is colder than *Gent et al.* [2011].

Because the reviewer has repeated the same comments with regards to our PI control simulation, we are obliged to address them again with the same arguments presented earlier:

1. The *Gent et al.* [2011] simulation was integrated for a total of only 1,300 model years whereas our PI control run has been integrated for a total of 5,170 years which is almost four times longer than the "benchmark simulation". There can be nothing sacred concerning the *Gent et al.* PI run as a CCSM4 "benchmark" for use in the context of PlioMIP. There is no physical basis on which the climatology of two simulations differing by almost 4,000 years in model years should agree.

2. Because our PI control is integrated for almost 4 times longer than that in *Gent et al.* [2011], it has resulted in a model that is much closer to equilibrium than the control of *Gent et al.*, which continued to lose heat at the top of the atmosphere (TOA) at a globally averaged rate of $-0.147\,Wm^{-2}$. This is larger than the $< |0.1|\,Wm^{-2}$ energy loss rate that *Gent et al.* consider desirable and larger than the $0.11\,Wm^{-2}$ energy imbalance that we get in our PI control $\mathrm{E^{280}P}$. Furthermore, the $\mathrm{E^{280}}$ model from which $\mathrm{E^{280}P}$ was branched off achieved an energy imbalance of $-0.02\,Wm^{-2}$ implying essentially perfect equilibrium. Continued integration of $\mathrm{E^{280}P}$ will lead to further reduction in the TOA energy imbalance.

3. The ocean model in *Gent et al.* [2011] CCSM4 PI simulation was configured with both (i) the overflow parametrization and (ii) tidal mixing scheme turned on. As we discuss in our manuscript, and which we further discuss below in response to the reviewer's concerns, these two options of the ocean model are tuned to the present day and as such should not be used for palaeo climate simulations [see *Peltier and Vettoretti* [2014], *Vettoretti and Peltier* [2013]]. But turning off these parameters in the Pliocene simulation, while keeping them active in the PI control would make the assessment of anomalies between the modern and the Pliocene simulation difficult due to this incompatibility between the two ocean models. It is for this reason that we have turned off both tidal mixing and overflow parametrization in our modern control, and therefore this is another difference between our PI control and *Gent et al.*.

4. There are some differences between the ocean mask in our PI control and that in *Gent et al.* [2011]. See *Vettoretti and Peltier* [2013] for details of the differences.

> P 1, Line 9-10: This is a different way in which the CCSM4 has been used with major changes to the ocean vertical mixing physics, so not just boundary conditions are different.

The reviewers comment here is with regards to the statement in the abstract "With the new boundary conditions, the CCSM4 model simulates a mid-Pliocene which is more than twice as warm as that with the boundary conditions used for PlioMIP Phase 1." The reviewer believes that this statement does not include some of the other changes (such as the change from the ocean mixing profile) that also distinguish the results of our simulation from those of *Rosenbloom*

*et al.* [2013]. With regards to this concern, we present the table below which compares the temperature anomaly of the Pliocene simulation with POP1 style profile of diapycnal diffusivity, namely, $\Delta TEoi^{400}{}_P$ with both controls (POP1 variants), as well as the anomaly of the 400 ppmv Pliocene simulation with the pelagic value of the diapycnal diffusivity ($\Delta TEoi^{400}$) with both controls (in this case the controls also employ the pelagic value of the diapycnal diffusivity). This comparison shows that the temperature anomalies for both variants of the Pliocene simulation are very similar and that the effect of the POP1 mixing profile is to increase the anomalies by 0.3 degrees. Both of the anomalies with respect to PI control (3.8°C and 3.5°C) are roughly twice as large as the anomaly simulated by *Rosenbloom et al.,* (who obtained an anomaly of 1.8°C) with the CCSM4 model and PlioMIP1 boundary conditions. This analysis demonstrates that the claim that with the new boundary conditions "the CCSM4 model simulates a mid-Pliocene which is more than twice as warm as that with the boundary conditions used for PlioMIP Phase 1" holds true despite the nature of the mixing profile. This is an important result on its own and discussion of it is included in our revised manuscript.

|  | Anomaly w.r.t 280 ppmv control | Anomaly w.r.t 400 ppmv control |
|---|---|---|
| $\Delta TEoi^{400}{}_P$ | 3.8 | 1.8 |
| $\Delta TEoi^{400}$ | 3.5 | 1.5 |

Since the reviewer has commented upon the impact of changes to the model, particularly in the ocean component, an additional point needs to be clarified. All our models have been run with the tidal mixing parametrization turned off. As per the discussion in our manuscript and in *Peltier and Vettoretti* [2014] this spatially heterogeneous field is tuned to the specific details of the present-day tidal regime and whose validity during time periods when the coastlines, bathymetric roughness, and eustatic sea level would have been different is questionable. Therefore, this parametrization has been turned off for the Pliocene simulations, and since the Pliocene simulations will need to be compared to the control simulations, in order to simplify this comparison without the subtlety of different mixing schemes, the parametrization has also been turned off in the controls. Our approach should be contrasted with that taken by *Rosenbloom et al.* [2013] with the CCSM4 model for PlioMIP. Although, the authors were careful to maintain consistency regarding the nature of mixing schemes between their control and Pliocene, they were nevertheless, forced to use the present day tidal mixing scheme within their Pliocene simulations because they opted to branch their Pliocene simulation from the existing PI control of *Gent et al.* [2011] which was configured with the present-day tidal mixing scheme.

> P 3, Lines 22-25: Cite Haywood et al (2011) when talking about the designs and aim of PlioMIP1

This reference will be included in the revised manuscript. Thank you for reminding us of this.

> P 6, lines 10-15: Please discuss the choice of the ocean convective parameterization. Was the KPP scheme used apart from turning off the tidal mixing component? Was tidal mixing also turned off when the Kv was kept constant? Perhaps show your preindustrial POP1 simulation's mixed layer depths. How is the 2-d distribution of mixed layer depth different

from the standard CCSM4 preindustrial simulation as described in Danabasoglu et al., J. Climate, 25, 2012?

The KPP convective parametrization was used in all simulations. The tidal mixing scheme was turned off in all simulations regardless of the choice of diapycnal diffusivity. The next figure compares our annual average mixed layer depth to that from *Danabasoglu et al.* [2012]. Our simulation is characterized by a robust mixed layer region in the north Atlantic and the GIN seas. The differences in the Labrador sea are expected based on the fact that overflow parametrization in our simulations was turned off (see *Danabasoglu et al.* [2012]).

[Figure]

P 7, line 8: The default 1° POP2 bathymetry field (KMT) is not from GTOPO30 as described here. The CCSM4 POP2 KMT field was generated from ETOPO2v2 as described and cited by Danabasoglu et al. 2012.

We thank the reviewer for bringing this to our attention. Comment to this effect has been included in the revised manuscript.

P 8, 18-20: This line, describing how the global mean salinity is adjusted for glacial climates to reflect a drop in sea level, and hence ocean volume, is not needed here as there is no glacial climate simulation. Thus, to make the manuscript more concise and focused, it should be deleted.

We believe that this statement, which clarifies the typical magnitude of ESL change for which a salinity adjustment becomes important, is pertinent to the discussion as to why a salinity adjustment was not performed and provides the necessary context for a reader who might be otherwise unfamiliar with the nature of such an adjustment.

P9, line 22: By what measure have the global mean SSTs come into equilibrium? Trends are still quite apparent in Fig. 3a.

The reviewer is, understandably, mistaken concerning an apparent SST trend. The reviewer is attempting to judge by eye the trend and there is a concavity in SST time series in the Pliocene simulations around year 1500. This is apparently giving the mistaken impression of a significant trend. However, numerical computation over the last two-hundred years of the simulation yields an SST trend of only $0.02°C/Century$ for E$oi^{400}$P, $0.04°C/Century$ for E$oi^{450}$P, $0.01°C/Century$ for E$^{280}$P and $0.01°C/Century$ for E$^{400}$P (also see table on the next page of this document). These values support our claim that the global mean SSTs have come into equilibrium, particularly for E$oi^{400}$P which is our "the Pliocene" simulation and for the control simulations. Because we now recognize that there is a possibility for a reader to mistakenly infer a larger trend than that which actually exists solely based on the figure, in the revised manuscript we now explicitly cite the numerically determined SST trends in the discussion of the equilibrium nature of our simulations.

P9, line 24: "...increases the rate of warming for the middle and lower and decreases the rate of warming for the upper ocean." This decrease/increase of trend is not apparent apart from the very beginning of the start of the POP1-Kv runs. The trends at the end of the POP1-Kv runs look very similar to the trends at the end of the POP2-Kv runs. Typically there is a flattening as the ocean model equilibrates, but this is not the case here. Comparing Figs 3b-d, there does seem to be a transfer of heat from the upper ocean at depths above 550 m which leads to cooling at the beginning of the POP1-Kv runs, while the deep ocean below 1850 m starts to warm. There was probably not a net heat loss to the atmosphere at the start through venting the upper ocean, otherwise the global mean temperature (Fig. 4) would show a cooling trend at this time instead of a warming trend.

We thank the reviewer for bringing to our attention an error in the text of the manuscript. What line 24 on page 9 should have said is: "This increases the rate of warming of the lower ocean and decreases the rate of warming for the upper and middle ocean." In the immediate centuries after the introduction of the POP1 profile of vertical mixing, there is vigorous vertical re-distribution of energy within the ocean. This leads to a cooling of the upper and the mid-depth oceans and warming of the abyssal ocean, which is expected on physical grounds. After this immediate response, the temperatures in the upper and mid-depth ocean continue to evolve with roughly the same trend as that before the introduction of the POP1 profile. In the lower ocean, however, the trends are characterized by a somewhat larger increase after the change.

P9, line 29: "...given the deep ocean sufficient time to come into equilibrium." Again, Fig 3d shows that the deep ocean is far from equilibrium with major drifts by the end of the simulations. P9, lines 30-32: Trends of global mean temperatures are twice as large O(.07K/century, shown in Fig. 2) as what is shown in Rosenbloom et al. 2013 for the CCSM4 PlioMIP1 simulation, which is around 0.03K/century. P13, lines 12-13: "...run to near statistical equilibrium..." By what measure are these simulations in a statistical equilibrium? As discussed above, deep ocean temperature trends are large and the global

mean ocean temperature at the end of the runs exceeds the CCSM4 PlioMIP1 simulation
discussed in Rosenbloom et al. 2013.

We expect that the trends of the global ocean temperature for different vertical levels are small enough that the simulations can be considered equilibrated. The table below shows the temperature trends in units of $°C/Century$ for different parts of the ocean computed over the last two hundred years. Note that the global ocean trends presented here for all simulations have been updated from a trend of 0.07°C/$Century$ for E$oi^{400}$P, 0.08°C/$Century$ for E$oi^{450}$P, 0.04°C/$Century$ for E$^{280}$P and 0.07°C/$Century$ for E$^{400}$P that was reported in the original manuscript. The updated trends will be incorporated in the revised manuscript, and we have also incorporated the table below into our manuscript. In this table the regions upper, middle and lower ocean are defined as in Figure 3 of the manuscript.

|  | Global Ocean | SSt | Upper Ocean | Middle Ocean | Lower Ocean |
|---|---|---|---|---|---|
| E$oi^{400}$P | 0.05 | 0.02 | 0.04 | 0.05 | 0.05 |
| E$oi^{450}$P | 0.05 | 0.04 | 0.03 | 0.05 | 0.06 |
| E$^{280}$P | 0.03 | 0.01 | 0.01 | 0.02 | 0.05 |
| E$^{400}$P | 0.06 | 0.01 | 0.03 | 0.05 | 0.07 |

Our work demonstrates the necessity of performing longer simulations with the revised PlioMIP2 boundary conditions than was necessary in the original PlioMIP, which did not include changes to the bathymetry and major oceanic gateways. Since no other group participating in PlioMIP2 has yet published results with the new boundary conditions fully implemented (*Kamae et al.* [2016] used the standard boundary conditions set which did not require changes to the land-sea mask) we suggest that our results, might be used by other groups to estimate how long they will need to run their own simulations to reach an adequate equilibrium.

The reviewer has correctly pointed out that the global mean temperature trend in *Rosenbloom et al.* [2013] Pliocene simulation was 0.03°C/$Century$ while that in our Pliocene simulation the trend is 0.07°C/$Century$ (older value from the manuscript, revised value in table above). At first glance, it may seem curious as to why the 500 year simulation of *Rosenbloom et al.* settled to a smaller ocean trend than our much longer simulation. However, there are several important reasons as to why their ocean model has reached equilibrium in such a short period of time:

1. The PRISM3 boundary conditions in PlioMIP did not specify any changes to the ocean bathymetry. This meant that the ocean in coupled-climate PlioMIP simulations that were initialized from present day temperature and salinity, or in models initialized from existing modern day runs (such as *Rosenbloom et al.*) were not characterized by the existence of a strong starting transient due to the modified bathymetry and circulation, or by the need to develop consistent T/S fields as required of the equilibrium state.

2. In addition to this, *Rosenbloom et al.* did not incorporate the PRISM3 changes to the land sea mask in the region of West Antarctica which is also expected to have aided in the faster approach to equilibrium.

3. Furthermore, the CCSM4 simulation of *Rosenbloom et al.* was branched at year 801 of the 1300 year PI control simulation. Given that their Pliocene simulation did not include

any changes to the bathymetry (which is partly because the PRISM3 dataset doesn't prescribe bathymetry, and partly because the authors did not implement the changes near Antarctica), save for the infilling of the shallow Hudson Bay region, this meant that each grid cell in the ocean was initialized with a robust and consistent temperature and salinity distribution, velocities, and circulation patterns. Including this 801 year PI simulation, their ocean model was actually integrated for 1300 years. In our case on the other hand, the changes that were necessary to apply the PlioMIP2 boundary conditions meant that several new ocean grid cells had to be created throughout the ocean, and therefore we've had to start our ocean from a state of rest (i.e. the velocities had to be completely spun up) and with interpolated temperature and salinity values for oceanic grid cells that exist on the Pliocene grid, but not on the modern day grid. This configuration requires longer integration to reach equilibrium.

Given the additional challenges that had to be overcome to get a fully PRISM4 compatible PlioMIP2 simulation going, it is remarkable that we have been able to get the trend in our Pliocene simulation down to $\sim 0.05°C/Century$.

P13, lines 18-34: Please indicate over what region of the water column and latitude range the maximum Atlantic Meridional Overturning Circulation metric was calculated. One needs to exclude the upper 500m or so of the wind-driven tropical cells.

The MOC maximum has indeed been computed while excluding the wind-driven tropical cell.

Figure 5 and Table 2 temperature both say Mean Annual Surface Air Temperature, but in comparing the same runs, the values do not agree. Table 2 numbers are higher than shown in Figure 5, suggesting that Table 2 numbers are TS, the Surface Temperature (radiative) which tends to be higher than the TREFHT or 2m air temperature typically used for 'MASAT'.

We thank the reviewer for pointing out this discrepancy. The values in Tables 2 and 3 are derived from the TREFHT field of the atmospheric component. These values are those that should be used for intercomparison with other studies. Due to technical issues that arose during the post processing of the original monthly history files that the model writes, the TREFHT variable did not get processed into time series for most models for the first several hundred years. Therefore, the bottom most hybrid level of the atmospheric temperature field was used to create the time series shown in manuscript Figure 5. The difference between the globally and yearly averaged values of these two variables only differs by a few tenths of a degree, while there is no change in the evolution of the time series. It is for this reason that Figure 5 shows the evolution of the globally and annually averaged temperature of the atmospheric layer closest to the ground. We ought to have made this clear but unfortunately this caveat didn't make it into the first draft. It has been made clear in the revised manuscript.

Figure 10: It is difficult to discern the contours in 10b-c. There appears to be possibly a very large cold bias in c and a smaller one in b, based on the minima shown in the color bars for these two panels of anomalies. There appears to be a large black swath in panel

> c emanating from the eastern boundary in the N. Pacific, suggesting a very large cold anomaly with respect to the Modern control.

The SST anomalies in manuscript Figure 10b and 10c show the Pliocene - control anomalies with respect to each of our controls. There is a region along the eastern boundary of the N. Pacific in the anomaly field with respect to 400 ppmv control over which the 400 ppmv Pliocene is cooler than 400 ppmv control. This was meant to have been clearer in the original figures, but the zero-anomaly contour line in our code was set to black which did not show up against the color scheme. We have clarified this with updated figures which show a zero-anomaly contour in white as shown below.

[Figure]

**References**

Bitz, C.M., Shell, K.M., Gent, P.R., Bailey, D.A., Danabasoglu, G., Armour, K.C., Holland, M.M., Kiehl, J.T., 2012. Climate Sensitivity of the Community Climate System Model, Version 4. J. Climate 25, 3053–3070. doi:10.1175/JCLI-D-11-00290.1

[revised manuscript text omitted]

---

## Author Comment (AC2) · 24 Apr 2017

**Response to Referee 2**

We are very grateful to the referee for the time taken to review our manuscript. The comments and suggestions have enabled us to further improve the clarity of the manuscript in a way that will make the work more accessible to the reader.

> In this paper, based on the PlioMIP2 boundary conditions, the authors use the CCSM4 to simulate the mid-Pliocene warm climate. The experiments are well designed and described in the paper. The paper is well written, and provide very good summaries for the Pliocene studies from aspects of ocean, land-ice reconstructions.
>
> However, in the paper, the authors do not clarify why they choose different diapycnal mixing k to carry out their experiments. In their models outputs, it is clear that the depth-dependent k remarkable changes the simulated ocean climate. Why do the authors only compare the simulated Pliocene and PI climate in the experiments with depth-dependent k? Do these experiments provide even stronger warming in the high- latitudes? In the revised version, I suggest the authors to plot figures similar to Figure 6, but based on the experiments with the constant k.

We thank the reviewer for asking us to clarify our reasons for choosing two types of $\kappa$. Although we have discussed in the manuscript our motivation behind the choice for $\kappa$ profiles that remove modern day constraints (page 6, last paragraph), we were remiss in not providing the same level of clarity regarding the decision to produce simulations based upon two different choices for $\kappa$. The primary motivation behind this choice was to study the sensitivity of the AMOC under PlioMIP2 boundary conditions for different choices of $\kappa$. This was motivated in part by the comparison of AMOC strengths for different PlioMIP simulations in *Zhang et al.* [2013]. We have added the following clarification in the Results and Discussion section of the revised manuscript:

> We mentioned previously that the simulations based upon the POP1 diapycnal diffusivity variant will constitute our primary PlioMIP2 results. This is simply because this depth dependent mixing profile has a structure that agrees with the diapycnal diffusivity of the deep ocean that is required to support the upwelling of deep abyssal water that is required to close the Antarctic bottom water cell. Therefore, in what follows, only the climatologies of the POP1 variants of the simulations will be discussed, and results from the constant $\kappa$ variants will only be discussed in very select situations.

The reviewer's comment concerning warming at high-latitudes led us to discover an interesting response of the mid-Pliocene surface atmosphere temperature to the choice of $\kappa$. The figure below is a modification of the manuscript Figure 8 such that in addition to the zonal-mean anomalies of $\mathrm{E}oi^{400}_\mathrm{P}$ compared to $\mathrm{E}^{280}_\mathrm{P}$ and $\mathrm{E}^{400}_\mathrm{P}$ (solid lines) it also shows the anomalies of $\mathrm{E}oi^{400}$ compared to $\mathrm{E}^{280}$ and $\mathrm{E}^{400}$ (dashed lines). This comparison shows an asymmetric response of the anomalies between the two hemispheres. The anomalies obtained with constant $\kappa$ simulations (dashed lines) are appreciably reduced in the southern hemisphere, compared to the anomalies obtained with POP1 type simulations (solid lines), while the opposite is true for the northern hemisphere. There are no appreciable differences throughout the tropical and extra-tropical latitudes. We believe this analysis provides a useful first estimate to the community regarding the magnitudes of changes that could be expected, and the regions where those changes can be expected with regards to changes in $\kappa$. This should assist the community to better understand the differences among results from the various models participating in PlioMIP2.

[Figure]

Furthermore, the differences in surface air temperature at high latitudes (originating from the choice of $\kappa$) is certainly going to be an important consideration for any future study that wishes to simulate the response and the stability of high-latitude Pliocene ice-sheets using ice-sheet models forced by simulated temperatures as boundary conditions. This analysis and accompanying discussion has been added to our revised manuscript.

> I also suggest the authors to plot AMOC for both two sets of experiments. It is helpful for other groups to judge which k provides better simulations for AMOC.

Figure 13 in the manuscript does show the AMOC for both two sets of experiments. There are two sets of curves in each sub-figure. The curves that begin mid-way in the sub-figures are for the POP1 type experiments. This is easier to see in the first sub-figure which shows the AMOC for the control experiment; there are two sets of red curves and green curves. In the other sub-figure, a discontinuity is present near year 1,100, but it is less apparent than in the control sub-figure primarily because the Pliocene constant $\kappa$ simulations were not continued for very long after branching.

> Since this work is taking part in the PlioMIP2, the authors should provide suggestions for other groups which set of experiments should be used in the future intercomparsions, the set with the constant k or the set with the depth-dependent k. Why?

We thank the reviewer for making an important point here. For PlioMIP2 we expect other groups to use the diagnostics computed with our POP1 type experiments. We have made this clearer in the revised manuscript in the section "Design of the Numerical Experiments" where we first introduce the two sets of experiments with the following comment:

> Since one of the defining large-scale characteristic of the global oceans are their vertical variation of the diapycnal diffusivity by an order of magnitude from low values in the thermocline to high values closer to the rough ocean bottom [*Waterhouse et al.*, 2014], our

POP1 variant simulations will constitute our primary PlioMIP2 simulations and it is these simulations that should be used by other groups for inter-comparison purposes.

**References**

Waterhouse, A.F., MacKinnon, J.A., Nash, J.D., Alford, M.H., Kunze, E., Simmons, H.L., Polzin, K.L., St Laurent, L.C., Sun, O.M., Pinkel, R., Talley, L.D., Whalen, C.B., Huussen, T.N., Carter, G.S., Fer, I., Waterman, S., Naveira Garabato, A.C., Sanford, T.B., Lee, C.M., 2014. Global Patterns of Diapycnal Mixing from Measurements of the Turbulent Dissipation Rate. J. Phys. Oceanogr. 44, 1854–1872. doi:10.1175/JPO-D-13-0104.1

Zhang, Z., Nisancioglu, K.H., Chandler, M.A., Haywood, A.M., Otto-Bliesner, B.L., Ramstein, G., Stepanek, C., Abe-Ouchi, A., Chan, W.-L., Bragg, F.J., Contoux, C., Dolan, A.M., Hill, D.J., Jost, A., Kamae, Y., Lohmann, G., Lunt, D.J., Rosenbloom, N.A., Sohl, L.E., Ueda, H., 2013. Mid-pliocene Atlantic Meridional Overturning Circulation not unlike modern. Clim. Past 9, 1495–1504. doi:10.5194/cp-9-1495-2013

---

## Author Response (AR1)

**Response to Editor**

We are very thankful to the Editor for taking the time to provide us with constructive comments that have allowed us, as we note below, to fix mistakes in our original submission and significantly improve our manuscript. We hope that our revised manuscript will have satisfied all of the Editor's remarks.

> I have studied the reviewers comments as well as the author response carefully. The paper presents a great deal of hard work and represents a very significant scientific contribution to PlioMIP2. It is clear that the paper will be worthy of publication in the PlioMIP2 CP special issue subject to the satisfactory completion of specific modifications/clarifications and additions. Whilst I totally understand the desire to robustly defend your own work, I would like to respectfully add that it is the authors responsibility to provide analyses and figures that satisfy the reviewers. This seems to have been a particularly important issue for this paper given the different results presented compared to the published and standard version of CCSM4 (by Rosenbloom et al. etc). I would also like to see the wording of the paper (in the title) altered to make it clear within the PlioMIP2 project that these results are derived from an altered version of CCSM4 (compared to the standard NCAR version). Perhaps something like "Regional and global climate for the mid-Pliocene using the University of Toronto version of CCSM4 and PlioMIP2 boundary conditions" would suffice.

We concur that the editor's suggested title "Regional and global climate for the mid-Pliocene using the University of Toronto version of CCSM4 and PlioMIP2 boundary conditions" would be a useful change to make. We have revised the manuscript to reflect this change.

> **Specific comments/requests.**
> **Pre-industrial control**
> Given the reviewers comments the pre-industrial control should be validated against observations, where available. For example, SST and sea ice versus HadiSST. The authors can refer to the Climate Data Guide to find relevant observational datasets (https://climatedataguide.ucar.edu/).
> A standard approach in climate modelling is that first a model simulation (PI or PD) should be validated against observations for the regions or variables being studied before analyzing sensitivity simulations, particularly in this study where it seems parameterizations have been modified, or turned off, in the PI control. I appreciate and understand the authors' comments and views on this matter but would also point out that the reasoning will not be intuitive to many that one should change or turn-off modern-based parameterizations as not appropriate for the Pliocene, but then also change or turn-off for the preindustrial where they are appropriate and well-tested in terms of enabling the model to perform well against observations. This discussion and relevant figures can be added as supplementary material of the paper and thus not disturbing the overall flow of the main paper.

We have added the requested comparisons of our pre-industrial control to the HadISST dataset in the Supplementary Materials. We have also clarified in the revised manuscript our reasons for switching off the overflow parameterization in the control simulations (revised manuscript page 5). With regards to the change to the diapycnal diffusivity parametrization, our original manuscript already discussed our reasons for changing the diapycnal diffusivity profile in both the control and the Pliocene simulations (revised manuscript page 6-7). In addition, in response to Referee 2's suggestion, we have expanded

that discussion to clarify for the readers which version of our experiments should be used for future intercomparisons with other PlioMIP/Pliocene studies (revised manuscript page 7).

**Pacific Warm Pool**

More justification for the expansion of the warm pool in the Eoi400 simulation seems to be required. Given the reviewer comments it appears to be important to distinguish whether the expanded warm pool is due to one or more of the new PlioMIP2 boundary conditions or rather the changes to the ocean parameterizations. Since the authors have been commendably comprehensive in performing the factorization simulations of PlioMIP2, they can provide a more detailed analysis to let the reader understand which of the PlioMIP2 forcings/boundary conditions (vegetation, different paleogeography, CO2) and/or the different POP parameters lead to this result. Note also that it is a little difficult to see an expansion of the warm pool in Fig. 10 from the panels given. Figure 10 could be redrafted to include contours every $\sim 1°C$, including negative anomalies in b) and c) and centered on the Pacific. Also, the large area of cooling in Eoi400 relative to the E400, which is somewhat obscured in Fig. 10c by showing only positive anomalies, needs some form of explanation.

To address the Editor's question "whether the expanded warm pool is due to one or more of the new PlioMIP2 boundary conditions or rather the changes to the ocean parameterizations" we introduce Figure 1 below, which compares the SST anomaly ($Eoi^{400}_P$ - $E^{280}_P$) to the SST anomaly ($Eoi^{400}$ - $E^{280}$) showing that the SST anomaly remains almost identical between these two choices of vastly different depth dependent profiles of the diapycnal diffusivities, $\kappa$. In particular, the regions of positive $1-2°C$ SST anomalies in the tropics and the sub tropics characterizing the expansion of the warm pool in the mid-Pliocene, is present in both anomaly plots. This comparison rules out any doubt that the expansion of the warm pool is the result of the modifications to the boundary conditions rather than from our choice of the $\kappa$ parameterization. Note however, this does not imply that the anomaly between "a Pliocene simulation" and "a PI" would be similar to that shown in Figure 1 for any/all choices of $\kappa$; indeed, for a spatially heterogeneous choice of $\kappa$, such as a tidal mixing scheme, the anomaly would be expected to be quite different in response to the large spatial variations in the strength of $\kappa$. What we have done by utilizing simplified schemes for $\kappa$, is to clearly isolate the impact of the revised boundary conditions from any subtleties associated with choices of $\kappa$. We thank the editor for prodding us to explore this question in more detail and helping us add another unique result to our paper. The arguments presented here have been incorporated into a section in the Supplementary Materials, and the main result referenced in the main manuscript.

Following the Editor's suggestion, we have also modified the anomaly plots in Figure 10 to better highlight the SST anomalies between the Pliocene and the two controls.

We have added a paragraph to the manuscript which provides a dynamical explanation for the large area of cooling seen in Figure 10c.

**Sea ice**

On page 4, of the authors' response, it is suggested that the sea ice climatology for the PI control period used in the Rosenbloom et al. 2013 PlioMIP1 comparison may have been different than for the period described in Gent et al. 2011. However, as reported in the atmospheric diagnostics on the publicly available web site: http://www.cesm.ucar.edu/experiments/cesm1.0/, the mean annual NH sea ice area is $11.73 \times 10^6 km^2$ for the period documented in Gent et al. 2011, which is approximately the same as shown in Table 2 of Rosenbloom et al. 2013.

I suggest comparing the sea ice predictions for the PI to the HadiSST preindustrial period. If the

[Figure]

Figure 1: SST anomalies of 400 ppmv Pliocene simulation with respect to PI control. (a) Anomaly with the POP1 variant simulations, (b) anomaly with the constant kappa set of simulations.

> sea ice extent is greater in this region compared to HadiSST this could, when compared to a Pliocene simulation without sea ice in that region, yield a large warm anomaly, which is just a function of a different sea ice result in the PI (compared to other PI CCSM4 simulations).
>
> On page 5 of the authors' response, the authors may be confusing the sea ice extent reported in Howell et al. 2016 with the sea ice area reported in Rosenbloom et al. 2013 for the CCSM4 PI. They show the annual cycle of sea ice area of their PI simulation (left), vs. Howell et al. 2016 (fig. 2) sea ice extent in the PlioMIP1 group (right). Sea ice extent will be larger than sea ice area. Sea Ice extent is computed by integrating over all grid boxes with sea ice, weighting by total area of the grid box, regardless of the fraction of ice within the grid box, thus obtaining an estimate of the size or extent of the ice pack. On the other hand sea ice area is computed by integrating area after weighting each grid box by the fraction covered by sea ice. Thus, the PI control mean monthly sea ice area climatology does not appear to be comparable to the sea ice extents reported in Howell et al. 2016. This should be re- examined and corrected as necessary.

We thank the editor for bringing to our attention the difference between 'sea ice area' and 'sea ice extent'. We were not familiar with this difference and were therefore treating them the same, which led us to incorrectly suggest in the response to Reviewer 1 that the climatological sea ice area in *Rosenbloom et al.* [2013] was much different from that in *Gent et al.* [2011]. The analysis in our manuscript was about the sea ice area, and our manuscript has been revised to remove any ambiguity.

We have also added comparison of the PI sea ice to HadISST in the supplementary materials.

> **North Atlantic SSTs**
> Otto-Bliesner et al., GRL, 2017 have shown that closing of the Arctic gateways improved the simulation of North Atlantic SSTs between 40-60N as compared to the earlier CCSM4 PlioMIP1 simulation. Please cite this paper.
>
> If the pre-industrial simulation (E280) in this paper does have more sea ice in the Labrador Sea in summer than might be indicated by HadiSST, this needs to be clearly stated when discussing warming in the North Atlantic, E400 - E280, Eoi400 - E280.
>
> Figure 11: Brierley et al. proxy-inferred SSTs are for early Pliocene. In fact in their Table S1, they include a temporal correction for those data that have the averaging interval centered in the mid-Pliocene in order to create their early Pliocene estimates. The PRISM3 data compilation needs to be used in this figure and only data points in the Brierley compilation that are in the PRISM3 time interval included since your paper is on the mid-Pliocene/PlioMIP2 paper.

We have cited and incorporated the findings of *Otto-Bliesner et al.* [2017] into the revised manuscript.

We have added a comment stating that the amplification of the SAT anomalies over the high-latitude North Atlantic and the surrounding regions is in part due to the significant difference in sea ice concentration between the mid-Pliocene and the PI control simulations.

Figure 11 has been revised to use the PRISM3 data points rather than *Brierley et al.* [2009] and the corresponding discussion text has been updated to reflect this change.

> **Winter temperature response over NH continents**
> The widespread cooling over the Northern Hemisphere continents during DJF is not supported by the proxy data. As such, more analysis of this feature needs to be included in the paper to understand such a response.
>
> Potential figures that could be referred to from the main text to the suppl section include maps of TLAI, snow cover, and circulation changes with the same NH projections as Figure 7. A comparison

to previous model results from PlioMIP1 may also help. The PlioMIP1 models with prescribed vegetation use the same vegetation reconstruction and similar Greenland ice sheet as PlioMIP2. At least some of the PlioMIP1 models show much reduced Arctic sea ice, which is relevant to the authors' hypothesis that the winter cooling is related to a reduced Arctic sea ice in the presented simulation. The authors can document which, if any, of the PlioMIP1 models also show widespread winter cooling over the NH continents.

We have expanded the discussion of the winter anomaly with a dynamical mechanism and literature references.

Regarding the Editor's comments about proxy data for the mid-Pliocene winter temperature patterns, we would be happy to include a comparison to a proxy dataset if the Editor could kindly refer us to one. We were hoping to find an appropriate dataset to use for the mid-Pliocene winter terrestrial temperature from the PlioMIP1 synthesis papers. Unless we have overlooked some literature, the only PlioMIP1 synthesis study examining data-model comparison for terrestrial surface air temperature is the recent study by *Salzmann et al.* [2013], but which deals exclusively with annual mean temperatures and does not appear to discuss any seasonal results. We have therefore added a new table to our manuscript where we show the remarkably good agreement between our predicted terrestrial annual mean temperatures, and those from the PRISM3 interval sites from *Salzmann et al.* [2013].

We have chosen not to include any analysis of the 2D temperature patterns from the original PlioMIP program because we are not currently set up to analyze those datasets and whose analysis will take considerable time. We also believe that this analysis would be more applicable to our second publication which will be an investigation of the sensitivity of the simulated mid-Pliocene to changes in various boundary conditions.

**Equilibrium Climate Sensitivity (ECS) and Earth System Sensitivity (ESS)**
It is very important to define what kind of sensitivity being examined very carefully. Table 2 in the paper presents the ECS (last column) of 7.4 and 6.3K. The manuscript also uses the term 'ECS' to discuss Table 2 on page 10. Is this instead ESS? If truly ECS, then these values are much greater than in the standard version of CCSM4. Comparing E400 to E280, one can estimate an ECS of $\sim 4K$, which is larger than the standard version of CCSM4 suggesting that the differences made to the ocean physics may have increased the ECS of this version of CCSM4. Perhaps it has also increased the ESS. Please clarify in the revised paper.

We thank the editor for catching a critical mistake in our manuscript. The sensitivity calculations reflect ESS and not ECS as the original manuscript (incorrectly) said. While we were clarifying a remark made by Referee 1 with regards to climate sensitivity, the fact that we had used an incorrect term in our manuscript escaped our attention. All instances referring to ECS have been replaced with ESS in the revised manuscript.

**Figure legends**
Please define details of averaging regions etc.

We have added the range of years over which the climatological averages were computed to relevant figure captions. This information has also been added to Table 2.

> **Spin up of the PI control**
> When asked for details about the pre-existing PI spin up of 3500 years length the authors reference Vettoretti and Peltier, 2013 (hereafter VP) on page 7 of the authors' response. However, VP describes a 3200 year long preindustrial spin up using CCSM3, not CCSM4. This detail of a change in model has not been included in the authors' response nor the manuscript. However, Fig. 2 in VP suggests the preindustrial spin up in VP may not be the one used here. Fig. 2c in VP, shows the evolution of sea ice area in the NH which is very smooth ending at year 3200 at $\sim 11 \times 10^6 km^2$. What is shown on pg. 7 of the authors' response, in contrast, is an evolution of NH sea ice 'extent' (is it 'extent' or area?) that shows an abrupt increase at about year 750 rising from $\sim 11.7 \times 10^6 km^2$ to $\sim 14 \times 10^6 km^2$, hence not looking like the smooth evolution in VP, fig. 2c. What happened at year 750 to cause the relatively abrupt increase in sea ice "extent" or area? Is the jump coincident with a growth of sea ice in the Labrador Sea? This then seems to persist into the later period of comparison. Another manuscript, Peltier and Vettoretti, GRL, 2014, hereafter PV, rather, uses CCSM4 for a long preindustrial control of length $\sim 2863$ years, still not 3500. Of the 2863 years, according to the PV paper, the last 1200 years had the tidal mixing and overflows turned off, POP1 Kv etc. like as described for the simulations occurring after the forks in Fig. 3-5. Did the authors go back and forth on the overflows and mixing schemes in the PI control as this suggests? Please give details of the spin up, and validation of the PI simulation over the period of comparison, either in the main text or Supplementary Information.

We thank the editor for bringing this matter to our attention and agree that a more detailed description of the PI control employed as the basis for our perturbation experiments is required. The PI control employed for the purpose of the PlioMIP2 analyses discussed in the present paper is unique to this paper but very closely related to the PI control employed for the purpose of the Dansgaard-Oeschger analyses discussed in *Peltier and Vettoretti* [2014] (henceforth PV14) and further in *Vettoretti and Peltier* [2015] (henceforth VP15). The PI control employed in PV14 was however created by branching from the NCAR PI control simulation at year 863 (simulation b40.1850.track1.1deg.006 run in the default CCSM4 configuration), and run for an additional 1,200 years with the overflow parameterization and the tidal mixing parametrization both turned off. This will be clear on the basis of Figure 2 in PV14 where the 1,200 years of this branched run are shown for the MOC strength time series in green.

For the purpose of the present paper we are employing a slightly different and unique PI control produced by a new 3,500 year run of the model under PI conditions (referred to as 'cesmpifv1mts'), from which the $E^{280}$ and the $E^{400}$ simulations in this paper were initialized. This control was produced by initializing in a similar manner to the glacial simulations discussed in *Vettoretti and Peltier* [2013] (henceforth VP13) and PV14, in which modern day temperature and salinity were assumed and the ocean and the atmosphere were assumed to be at rest. The model was then run continuously for 3,500 years with the overflow parameterization and the tidal mixing parametrization turned off throughout the duration of the simulation. Because we were concerned that the PI control employed in PV14 could have inherited memory of branching from the standard configuration of CCSM4, we considered it wise to produce an entirely new PI control. In retrospect this PI control is found to be very close to that employed in PV14. Another slight difference between these distinct PI controls (a technical difference essentially irrelevant to the simulated climate) is that the modern-day topography and bathymetry for cesmpifv1mts was generated in-house from ETOPO1 (as mentioned in the main text of the manuscript), whereas the PI control in PV14 used the default NCAR generated modern-day boundary conditions based upon ETOPO2v2, because that simulation was branched from the NCAR PI control as mentioned above. Since cesmpifv1mts has not been discussed previously in the literature, we have provided detailed spin up diagnostics for this simulation in section 2 of the supplementary materials.

The AMOC in the PI simulation employed in PV14 is ~22 Sv at equilibrium as will be clear on the

basis of Figure 2 of that paper. Cesmpifv1mts was also on track to settle into a stable AMOC of ∼22 Sv, before apparently passing a weak "thermal threshold", not unlike that seen under glacial conditions using CCSM3 (VP13) and CCSM4 (PV14) across which there is a sudden reduction in AMOC strength. It is likely that this nonlinear thermal threshold was not encountered in the PV14 control simulation because it was (i) not run long enough, or (ii) because of its unique simulation history (NCAR run followed by an in-house extension of the run with different parameterizations). Nevertheless this threshold effect has only a very modest impact on the eventual statistical equilibrium. The cesmpifv1mts simulation, along with the results from VP13 and PV14, demonstrate that it is generally best practice to run a simulation for many thousands of years so as to achieve an equilibrium state, if indeed one can be expected to exist for the initial and boundary conditions being employed.

**References**

[revised manuscript text omitted]

~~The globally averaged MASAT is $16.8\,°C$ for model $\mathrm{E}oi^{400}\mathrm{P}$ and $17.3\,°C$ for model $\mathrm{E}oi^{450}\mathrm{P}$ (Table 5). The 400 ppmv mid-Pliocene simulation is $3.8\,°C$ warmer than the PI, and the 450 ppmv mid-Pliocene simulation is $4.3\,°C$ warmer. These values are larger than the magnitude of the anomaly predicted by every model that has been exercised previously in the context of the PlioMIP programand double the anomaly thatfound ($1.86\,°C$) using CCSM4. Our simulated anomaly is also much higher than that found by($2.4\,°C$) using the PlioMIP2 boundary conditions and the MRI-CGGM2.3 coupled climate model. A likely explanation as to why their anomaly is lower than ours could be due to their choice of the 'standard' boundary conditions set which does not require changes to the land-sea mask in the model. A more likely explanation of the difference, however, is simply that the relatively short integration length of 500 years would not have been sufficient to enable the ocean in their model from reaching a state of quasi-equilibrium.~~

Table 4 compares the MASAT simulated in the $\mathrm{E}oi^{400}\mathrm{P}$ experiment to the PRISM3 interval (3.3 - 3.0 Mya) MASATs, which were used in the original PlioMIP program to compare the simulated temperatures to proxy based inferences of terrestrial temperatures (*Salzmann et al.,* 2013). A caveat that must be kept in mind when comparing our simulated results to this compilation is that the PRISM3 proxy estimate reflects an average over the long PRISM3 time interval relevant to PlioMIP, whereas for PlioMIP2 the focus has shifted onto a specific interglacial. Subject to this caveat, it is observed that except for the two Siberian sites (Chara Basin and Lake Baikal) our simulated MASATs are in very good agreement with proxy inferences.

The globally averaged MASAT is $16.8\,°C$ for model $\mathrm{E}oi^{400}\mathrm{P}$ and $17.3\,°C$ for model $\mathrm{E}oi^{450}\mathrm{
[revised manuscript text omitted]